# Inner lumen proteins stabilize doublet microtubules in cilia and flagella

Mikito Owa[1], Takayuki Uchihashi [2], Haru-aki Yanagisawa[1], Takashi Yamano[3], Hiro Iguchi[3], Hideya Fukuzawa[3], Ken-ichi Wakabayashi[4], Toshio Ando [5] & Masahide Kikkawa [1]

Motile cilia are microtubule-based organelles that play important roles in most eukaryotes. Although axonemal microtubules are sufficiently stable to withstand their beating motion, it remains unknown how they are stabilized while serving as tracks for axonemal dyneins. To address this question, we have identified two uncharacterized proteins, FAP45 and FAP52, as microtubule inner proteins (MIPs) in *Chlamydomonas*. These proteins are conserved among eukaryotes with motile cilia. Using cryo-electron tomography (cryo-ET) and high-speed atomic force microscopy (HS-AFM), we show that lack of these proteins leads to a loss of inner protrusions in B-tubules and less stable microtubules. These protrusions are located near the inner junctions of doublet microtubules and lack of both FAP52 and a known inner junction protein FAP20 results in detachment of the B-tubule from the A-tubule, as well as flagellar shortening. These results demonstrate that FAP45 and FAP52 bind to the inside of microtubules and stabilize ciliary axonemes.

[1] Department of Cell Biology and Anatomy, Graduate School of Medicine, The University of Tokyo, 7-3-1 Hongo Bunkyo-ku, Tokyo 113-0033, Japan. [2] Department of Physics and Structural Biology Research Center, Chikusa-ku, Nagoya University, Nagoya 464-8602, Japan. [3] Graduate School of Biostudies, Kyoto University, Kyoto 606-8502, Japan. [4] Laboratory for Chemistry and Life Science, Institute of Innovative Research, Tokyo Institute of Technology, Yokohama 226-8503, Japan. [5] WPI Nano Life Science Institute, Kanazawa University, Kakuma, Kanazawa 920-1192, Japan. Correspondence and requests for materials should be addressed to M.K. (email: mkikkawa@m.u-tokyo.ac.jp)

Cilia and flagella are microtubule-based organelles that operate as both antennae and propellers in eukaryotic cells. Cilia are classified as either non-motile or motile. Non-motile cilia, or primary cilia, function as antennae and are involved in signal transduction through the hedgehog, Wnt, and $Ca^{2+}$ signaling[1]. Motile cilia and flagella beat at 20–60 Hz[2,3] and drive cellular motility and fluid flow. Since the cells in most tissues are ciliated, ciliary dysfunction leads to various types of human diseases termed ciliopathies, which include hydrocephalus, *situs inversus*, retinal degeneration, and nephronophthisis[4]. Furthermore, recent studies revealed that the assembly of cilia is significantly decreased in several types of tumors[5,6], implying some correlation between ciliogenesis and tumorigenesis.

The structure and protein composition of motile cilia and flagella are well conserved among eukaryotes. The axoneme, the core structure of cilia and flagella, is composed of a central pair of microtubules cylindrically surrounded by nine doublet microtubules (DMTs). This arrangement is often referred to as the 9 +2 structure (Fig. 1a). In DMTs, 10 protofilaments of the B-tubule are attached to 13 protofilaments of the A-tubule at the inner and outer junction (Fig. 1a). In motile cilia and flagella, structures essential for motility, such as axonemal dyneins, radial spokes, and the nexin–dynein regulatory complex (N-DRC), are arranged on DMTs with a 96-nm repeating unit[7–9]. Axonemal dyneins bound on A-tubules slide on the neighboring B-tubules and this sliding propagates along the axoneme, generating a bending force. The activity of dyneins is regulated by the interaction between the radial spokes and the central pair of singlet MTs[10–12]. The N-DRC forms a crossbridge between neighboring DMTs and is required for orchestrating dynein activity and axonemal integrity[13–16].

Recent advances in cryo-electron tomography (cryo-ET) techniques have revealed structural details of DMTs, yet a significant question remains—what stabilizes DMTs? Cytoplasmic MTs frequently bend with various degrees of curvature[17,18] and this bending occasionally induces MT breakage[19]. Furthermore, a recent study demonstrated that cytoplasmic MTs were damaged even after bending only several times[20]. Therefore, to ensure a high beat frequency, the DMTs of motile cilia and flagella should be structurally robust compared with cytoplasmic MTs. However, the mechanism that provides DMTs with such robustness remains to be clarified.

Nicastro and colleagues have provided insights into the structural basis of DMTs using cryo-ET of *Chlamydomonas* flagella. They reported periodic high densities on the inner surfaces of A-tubules and B-tubules, which they named microtubule inner proteins (MIPs, Fig. 1a)[7,21]. To date, two MIPs in the A-tubule have been identified by recent studies[22,23]. MIPs have also been observed in the axonemes of higher organisms[7,21,24–26], implying that these inner structures are essential for the integrity of DMTs in motile cilia and flagella.

In this paper, we explore the mechanisms stabilizing DMTs using *Chlamydomonas* mutants and various techniques, including cryo-ET and high-speed atomic force microscopy (HS-AFM). We find that two uncharacterized flagellar-associated proteins (FAP), FAP45 and FAP52, are essential for the stability of B-tubules. Lack of both FAP45 and FAP52 leads to the loss of MIPs in B-tubules, resulting in disruption of the MT walls. Furthermore, the B-tubule wall detaches from the A-tubule at the inner junction, and flagellar shortening is observed when both FAP52 and FAP20 are absent. These results indicate that the B-tubule is reinforced by MIPs and its stability is essential for maintaining the structure of the axoneme.

## Results

### Identification of proteins essential for DMT stabilization.
To identify proteins that stabilize DMTs, we searched the *Chlamydomonas* flagellar proteome database[27]. We postulated that those proteins are (1) abundant in the axoneme fraction, (2) tightly bound on the axoneme even in the presence of high salt, and (3) highly conserved among ciliated organisms. The proteins that satisfy these criteria are listed in Supplementary Table 1. We focused on FAP45, an uncharacterized protein whose peptides were most frequently found in the proteomic analysis (Supplementary Table 1, total unique peptide and Axo columns). FAP45 is a coiled-coil protein composed of 501 amino acids, has a predicted molecular weight of ~59 kDa, and is conserved among organisms with motile cilia. The human ortholog of FAP45 is coiled-coil domain-containing protein 19 (CCDC19), also known as NESG1. The transcript of this protein is enriched in the nasopharyngeal epithelium and trachea[28]. However, the functions of FAP45/CCDC19 are unclear. Thus, we chose FAP45 as a candidate of proteins stabilizing DMTs.

We first looked for partner(s) that interacts with FAP45 on the axoneme using chemical crosslinking. We treated isolated wild type axonemes with 1-ethyl-3-(3-dimethylaminopropyl)carbodiimide (EDC, zero-length crosslinker). The crosslinked products were solubilized and immunoprecipitated with a polyclonal anti-FAP45 antibody. Tubulin was identified as the major crosslinked partner of FAP45 using western blot (Fig. 1c, open arrowhead), suggesting a direct interaction between FAP45 and tubulin. In addition, a mass spectrometric analysis revealed that the ~130 kDa product is composed of FAP45 (~59 kDa) and FAP52 (~66 kDa), probably in a 1:1 ratio based on the molecular weight (Fig. 1c and d, filled arrowhead; Supplementary Fig. 1d, arrowhead; Supplementary Table 2).

FAP52 is the *Chlamydomonas* ortholog of human WD40 repeat domain 16 (*WDR16*) and is listed in Supplementary Table 1. It was previously shown that *WDR16* knockdown in zebrafish led to severe hydrocephalus[29]. Furthermore, a recent study reported that homozygous deletion of the *WDR16* gene in human leads to situs anomalies, which are typical phenotypes of ciliopathy[30]. However, the function of the FAP52/WDR16 protein in flagellar motility and axonemal structure remains unclear. We hypothesized that FAP45 and FAP52 proteins play a role in stabilizing DMTs.

### Characterization of *Chlamydomonas* FAP45 and FAP52 mutants.
We used *Chlamydomonas* mutants lacking FAP45 and FAP52 to investigate their function. The mutants were isolated from ~10,000 clones in an insertionally mutagenized *Chlamydomonas* library[31]. In these mutant strains, the ~1.8 kbp *aphVIII* fragment was inserted into the 5′UTR of the FAP45 gene or the first exon of the FAP52 gene (Supplementary Fig. 1b). By a southern blot, we confirmed that each mutant has one *aphVIII* fragment inserted only at the targeted locus (Supplementary Fig. 1c). These strains did not express detectable FAP45 or FAP52 protein as analyzed by a western blot of axonemal fractions and immunofluorescence of the axonemes (Fig. 1b; Supplementary Fig. 2b and c). On the other hand, *fap45* and *fap52* axonemes retained known major axonemal components (Fig. 1b). Also, FAP45 and FAP52 were incorporated into the axoneme independently of each other (Fig. 1b). The ~130 kDa crosslinked product of FAP45 and FAP52 was detected in the wild type axoneme treated with EDC but not in the crosslinked *fap45* or *fap52* axoneme (Fig. 1c and d, filled arrowheads), consistent with the mass spectroscopic analysis.

Next, we examined the motility phenotype of *fap45* and *fap52*. The swimming velocity and beat frequency of *fap45* cells were slightly reduced (Fig. 1e; Supplementary Movie 1), whereas the phenotype of *fap52* was quite similar to that of wild type (Fig. 1f; Supplementary Movie 1). Intriguingly, the double mutant *fap45*

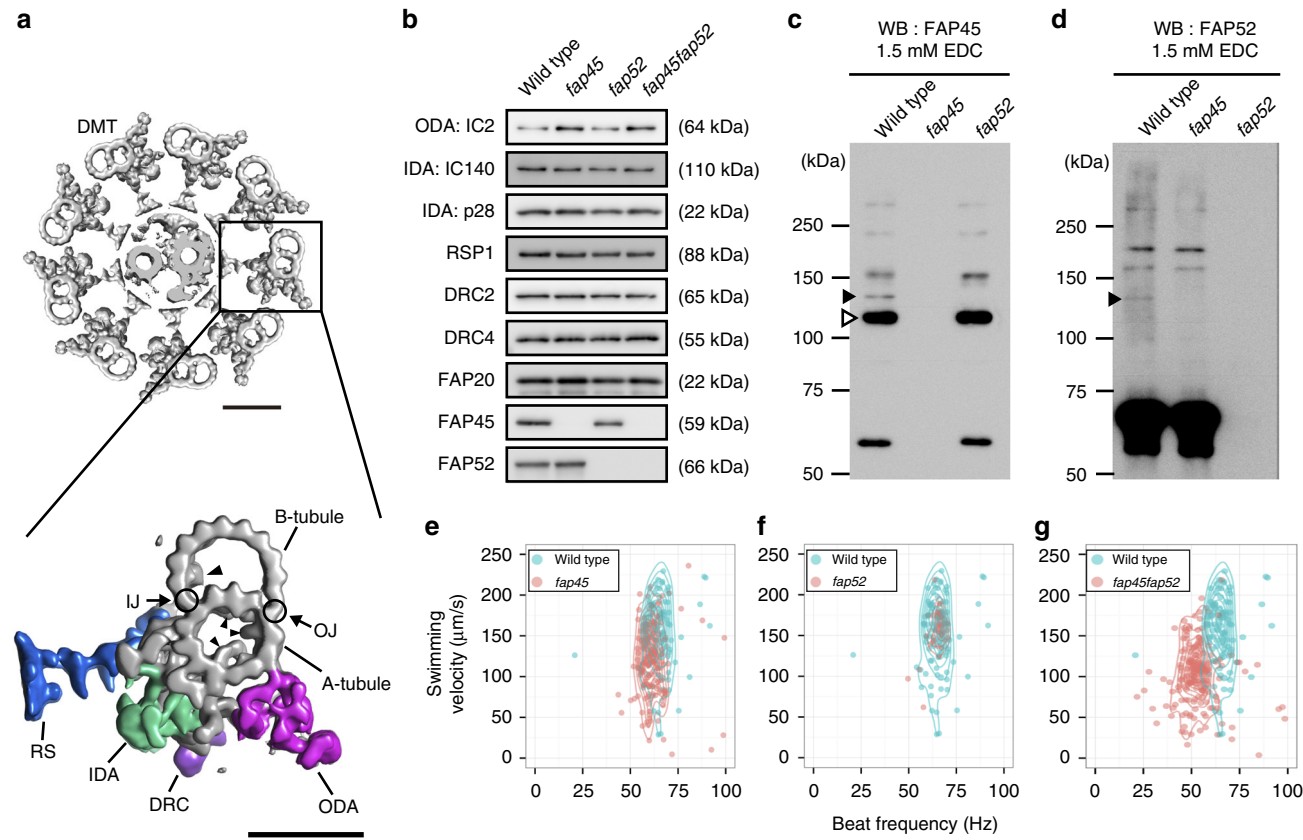

**Fig. 1** Characteristics of the FAP45 and FAP52 null mutants. **a** Top: a transverse view of the 9+2 structure of the *Chlamydomonas* axoneme. Scale bar = 50 nm. Bottom: a magnified DMT. Arrowheads indicate MIPs. Major structures are colored; Outer dynein arm (ODA, magenta); Inner dynein arm (IDA, green); Dynein regulatory complex (DRC, purple); Radial spoke (RS, blue). IJ inner junction, OJ outer junction. Scale bar = 25 nm. **b** Western blot analyses of wild type, *fap45*, *fap52*, and *fap45fap52* double mutant axonemes stained with various antibodies. FAP45 and FAP52 proteins were not detected in *fap45* and *fap52*, respectively. Proteins essential for flagellar motility (ODA-IC2: outer dynein arm-intermediate chain 2; IDA-IC140: inner dynein arm-intermediate chain 140; IDA-p28: inner dynein arm-light chain p28; RSP1: radial spoke protein 1; DRC2 and 4: dynein regulatory complex 2 and 4; FAP20: inner junction protein of DMT) were not reduced in the mutants. **c**, **d** Axonemes from wild type and mutant *Chlamydomonas* crosslinked using EDC (zero-length crosslinker) were immunoblotted with anti-FAP45 (**c**) and anti-FAP52 antibodies (**d**). Filled arrowheads indicate the crosslinked product of FAP45 and FAP52 in a 1:1 ratio. The open arrowhead indicates the crosslinked product of FAP45 and tubulin. **e**–**g** Motility phenotypes of the mutants were assayed using the CLONA system. Swimming velocity and beat frequency were slightly reduced in *fap45*, whereas no significant reduction was observed in *fap52*. *fap45fap52* showed a more severe phenotype than did *fap45*

and *fap52* (*fap45fap52*) swam significantly more slowly than wild type, with a low beat frequency (Fig. 1g; Supplementary Movie 1). Flagellar beating of swimming cells in these mutants appeared normal (Supplementary Movie 2). Furthermore, there was an accumulation of non-motile *fap45fap52* cells under confluent culture conditions (Supplementary Movie 3). We also observed the swimming behavior of wild type and *fap45fap52* log phase cells in the viscous medium. Most of the wild type cells swam slowly but smoothly (Supplementary Movie 4) whereas many of the *fap45fap52* cells stopped swimming (67 of 78 cells) or struggled to swim against the viscous load (Supplementary Fig. 3; Supplementary Movie 4), indicating that *fap45fap52* flagella cannot beat properly under this condition. These results indicate that the lack of both FAP45 and FAP52 led to synergistic effects on motility phenotypes.

**FAP45 and FAP52 are luminal proteins in B-tubules**. We biochemically localized FAP45 and FAP52 in DMTs by fractionating the axonemal proteins using increasing concentrations of sarkosyl[32] (Supplementary Fig. 4a). FAP45 began to be extracted from the pellet at 0.2% sarkosyl, half of the FAP45 was extracted

at 0.3%, and the protein was completely extracted from the pellet at 0.7% (Supplementary Fig. 4a). FAP52 started to be solubilized at 0.3% sarkosyl and almost all the protein was solubilized at 0.7%, although a small amount of protein still remained in the pellet. These behaviors correlate well with solubilization of the B-tubule.

To distinguish whether FAP45 and FAP52 are located outside or inside the B-tubule, we prepared axonemes containing biotinylated FAPs and tested whether or not the proteins are accessible to streptavidin. These axonemes were purified from rescue strains expressing FAP proteins whose N-terminus or C-terminus was fused to biotin carboxyl carrier protein tag (BCCP tag)[33,34]. These rescued strains expressing BCCP-tagged proteins (Supplementary Fig. 4b and c) swam like wild type. However, essentially no signal of biotinylation was detected on those axonemes (Supplementary Fig. 4d), suggesting that streptavidin access to BCCP tag was prevented, presumably by the B-tubule wall. Therefore, we treated axonemes with 0.15% sarkosyl, which partially broke the B-tubule wall and, as expected, resulted in detection of the tagged proteins (Supplementary Fig. 4e). Taken together with the results that the loss of FAP45 and FAP52 did not affect the incorporation of major axonemal proteins bound

on the outer surface of the DMT (Fig. 1b), these data indicate that FAP45 and FAP52 are most likely luminal proteins enclosed by the B-tubule.

**FAP45 and FAP52 form MIPs in B-tubules**. To reveal the defect caused by the loss of FAP45 and FAP52, we observed the mutant axonemes by cryo-ET. We found that the density of dynein b, a species of inner arm dyneins was decreased in *fap45* and *fap52* (Fig. 2a–c). This result suggests that in the mutants the number of dynein b may be reduced or dynein b cannot bind to the B-tubule properly for some reason.

For further visualization of missing structures in *fap45* and *fap52*, we applied Student's *t*-tests to compare wild type and mutant density maps[33] (Fig. 3e and f, right). In agreement with the above results, the mutants also showed structural defects inside of the B-tubule (Fig. 3).

A comparison between wild type and *fap45* DMTs shows that the luminal surface of wild type B-tubule is smoother than that of *fap45* (Fig. 3d and e). In the *fap45* DMTs, the grooves between B-tubule protofilaments B7-B9 are prominent (Fig. 3b and e; Supplementary Fig. 5b). The grooves seem to be partially filled with filamentous densities as shown in the t-map (Fig. 3e right, highlighted in green). These densities are arranged in a 48-nm repeat, and reminiscent of a lateral filament connecting MIP3, fMIP B8B9, and fMIP B7B8 in *Tetrahymena* DMTs[35]. Therefore, we named the missing structures as MIP3c (Fig. 3e right, circled in red).

A comparison between the wild type and *fap52* DMT revealed that an arch-like density is missing in the B-tubule of the *fap52* mutant. In the wild type DMT, there are a large arch-like density and a small spike-shaped density in a 16-nm repeat inside of the B-tubule on the protofilament B9 and B10, called MIP3a and MIP3b, respectively (Fig. 3a and d, highlighted in yellow and orange; Supplementary Fig. 5a)[21]. The DMTs of *fap52* completely lacked the densities of MIP3a, whereas MIP3b appear to be unchanged (Fig. 3c and f; Supplementary Fig. 5a).

Next, we tested whether the predicted atomic structures of FAP45 and FAP52 can fit into these densities. *Chlamydomonas* FAP52 contains 11 WD40 domains (Supplementary Fig. 1a). Since WD40 repeat domains typically form a seven-bladed β-propeller, FAP52 probably contains two β-propellers. Indeed, a model comprising two β-propellers fit well into the *Chlamydomonas* density map (Supplementary Fig. 5d), as well as the recently published sub-nanometer map of *Tetrahymena*[35] (Supplementary Fig. 5e), suggesting that the density of MIP3a corresponds to one molecule of FAP52. For MIP3c, predicted coiled-coils of FAP45 protein were fit into the density (Supplementary Fig. 5c). FAP45 is mainly composed of helices and coiled-coils that account for 85% of the total sequence (Supplementary Fig. 1a). The total length of helices and coiled coils in FAP45 is ~60 nm. Therefore, the circled densities in Supplementary Fig. 5c are probable one unit of MIP3c. Thus, MIP3c probably consists of FAP45.

To investigate the reasons why the double mutant *fap45fap52* cells swim more slowly than single mutants, we observed the *fap45fap52* axoneme using cryo-ET. In this case, we could not apply 3D sub-tomographic averaging because many of the axonemes were frayed and the B-tubules were partially missing (Fig. 4a and b). Since cryo-ET is not suitable for observing the shapes of individual DMTs due to the missing wedge effects, thin sections of the *fap45fap52* axoneme were observed by conventional EM and revealed that the B-tubules were missing in ~33.8% of the outer DMTs (Fig. 4c). Among them, 23% lacked all of the protofilaments in the B-tubules, whereas 77% retained several protofilaments. These data suggest that the loss of both

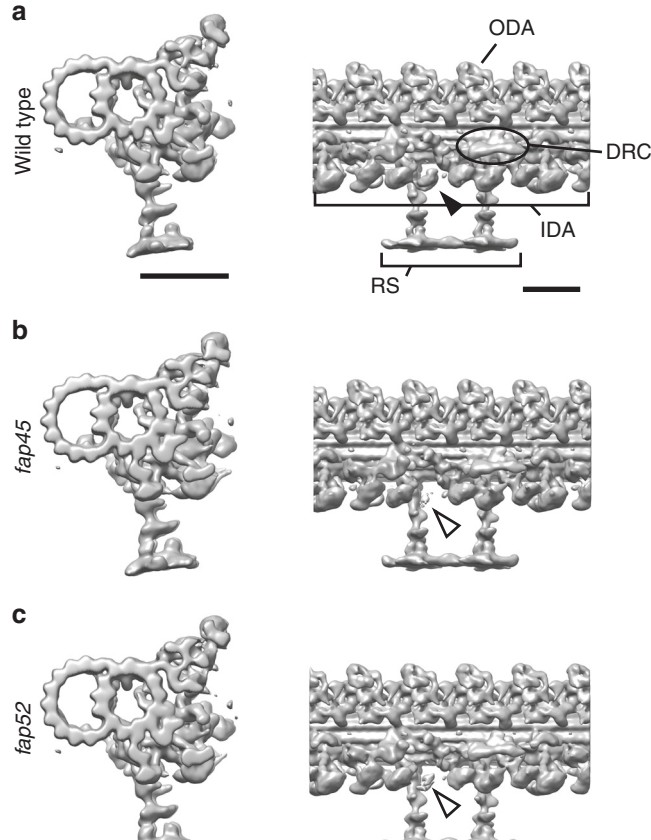

**Fig. 2** Dynein b density is reduced in the *fap45* and *fap52* DMT. Isosurface renderings of averaged DMT repeats: wild type (**a**), *fap45* (**b**), and *fap52* (**c**). The density of dynein b is decreased in *fap45* and *fap52* (arrowheads). Other species of IDAs, ODAs, RS, and N-DRC were properly arranged on the DMT in *fap45* and *fap52*. Scale bar = 25 nm

FAP45 and FAP52 decreases structural stability between the B-tubule protofilaments.

**Direct observation of B-tubule depolymerization by HS-AFM**. The above static DMT structures suggested that the B-tubules of *fap45fap52* are less stable than those of wild type. We directly observed the stability of the B-tubule using HS-AFM[36], in which a small cantilever tip (tip diameter ~1–2 nm) intermittently taps the DMT surface fixed on a mica stage. This intermittent contact can eliminate frictional force during lateral scanning and damage to the DMT caused by scanning because the feedback operation maintains a constant tip–sample interaction force and the force is applied for a short time (<100 ns).

We first observed in vitro polymerized MTs in the absence of taxol, a stabilizing agent. It was previously reported[25] that MTs depolymerize spontaneously within two or three frames (1.0–1.5 s), suggesting that depolymerization requires <50 nm/s. Actual depolymerization could be faster[26] but our HS-AFM has insufficient temporal resolution to verify this. Compared with in vitro polymerized MTs, DMTs are more stable and do not depolymerize spontaneously under normal HS-AFM conditions and their structures can be observed in detail.

When DMTs are adsorbed on the mica surface, AFM images were classified into three types as shown in Supplementary Fig. 7; Class 1: The 24 nm periodicity of ODAs was visualized on the top of the DMT, suggesting that the B-tubule and the radial spokes were immobilized on the mica; Class 2: The heads of the radial spokes in 96 nm periodicity were visualized on the top of the DMT,

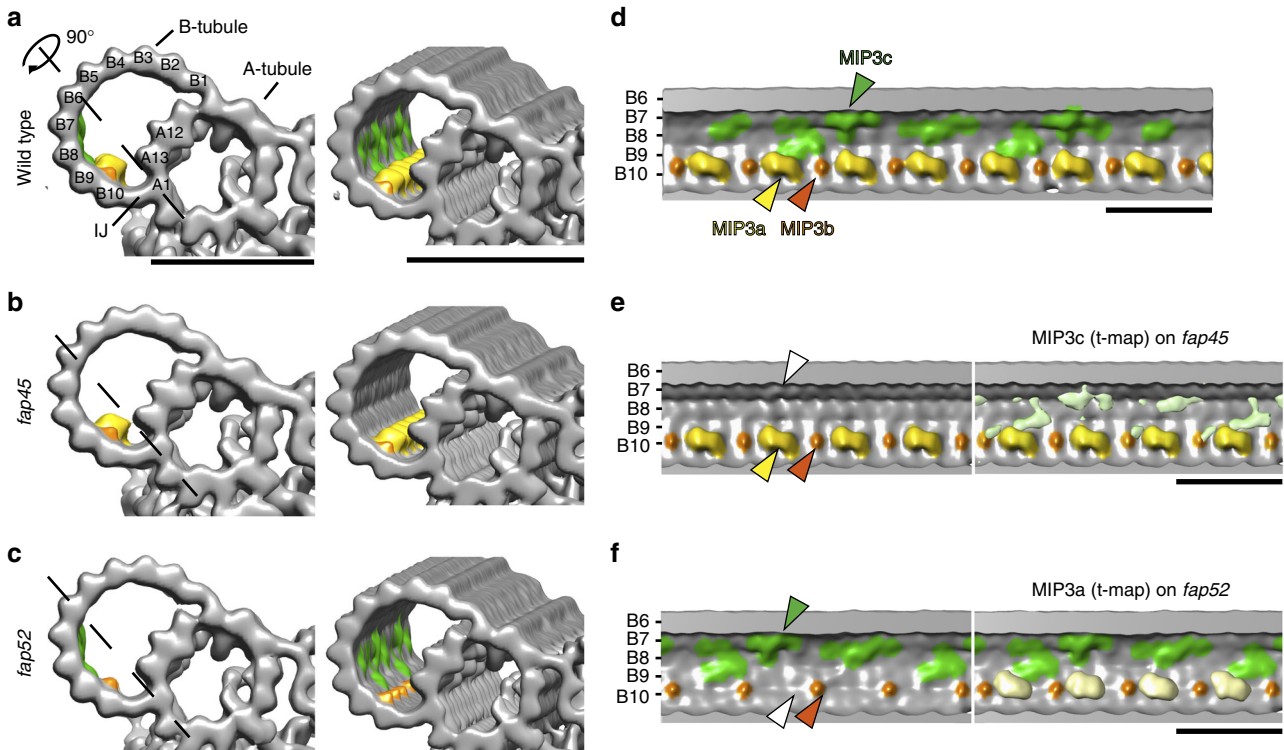

**Fig. 3** *fap45* and *fap52* mutant have structural defects in B-tubules. Isosurface renderings of averaged axonemal 96-nm repeats from wild type (**a**, **d**), *fap45* (**b**, **e**), and *fap52* (**c**, **f**). (**a–c**) are cross-sectional and oblique, and (**d–f**) are longitudinal views. The black dashed lines in (**a–c**) indicate the slicing plane of (**d–f**). Protofilament numbers are labeled in (**a**) and (**d–f**). Missing structures in *fap45* and *fap52* were determined by Student's *t*-tests as previously described[33]. The *t*-value maps acquired by the *t*-tests were overlaid on the renderings of *fap45* (**e**, light green) and *fap52* (**f**, light yellow). In this figure, MIP3a and b were colored by yellow and orange, respectively. MIP3c (the missing structure in *fap45*) was colored by green, according to the *t*-value map shown in (**e**). Scale bar = 25 nm

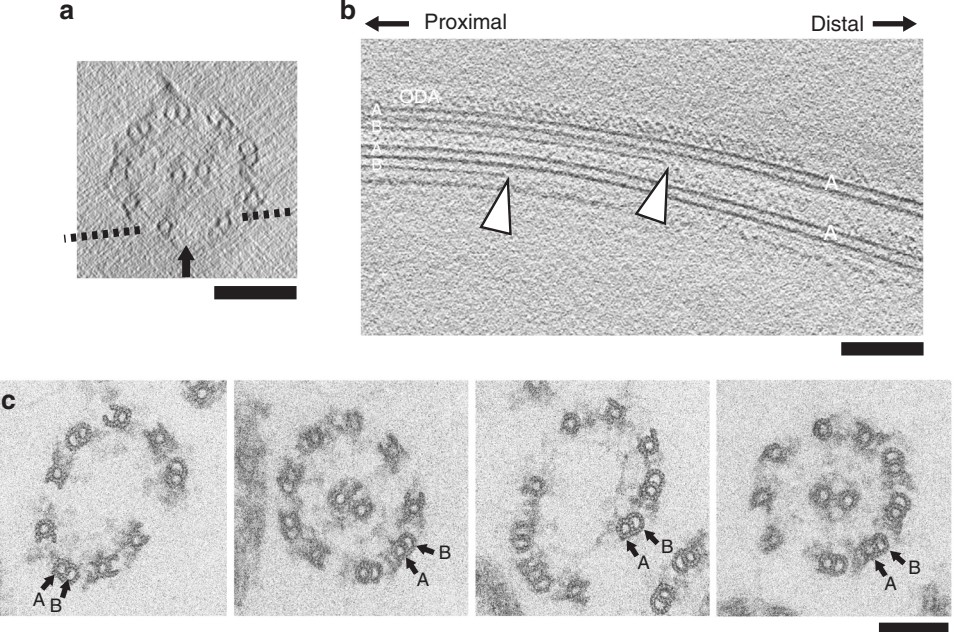

**Fig. 4** B-tubules are partially depolymerized in *fap45fap52*. **a**, **b** A typical tomogram of a *fap45fap52* axoneme in cross-sectional (**a**) and longitudinal view (**b**). A dashed line in (**a**) indicates the slicing plane of (**b**). Arrowheads in (**b**) show break points in the B-tubules. **c** Typical thin section TEM images of *fap45fap52* axonemes. Broken B-tubules were observed in 33.8% of the DMTs (*n* = 552). The 9+2 structure was disorganized and the axonemes were frayed in regions containing broken B-tubules. Scale bar = 100 nm

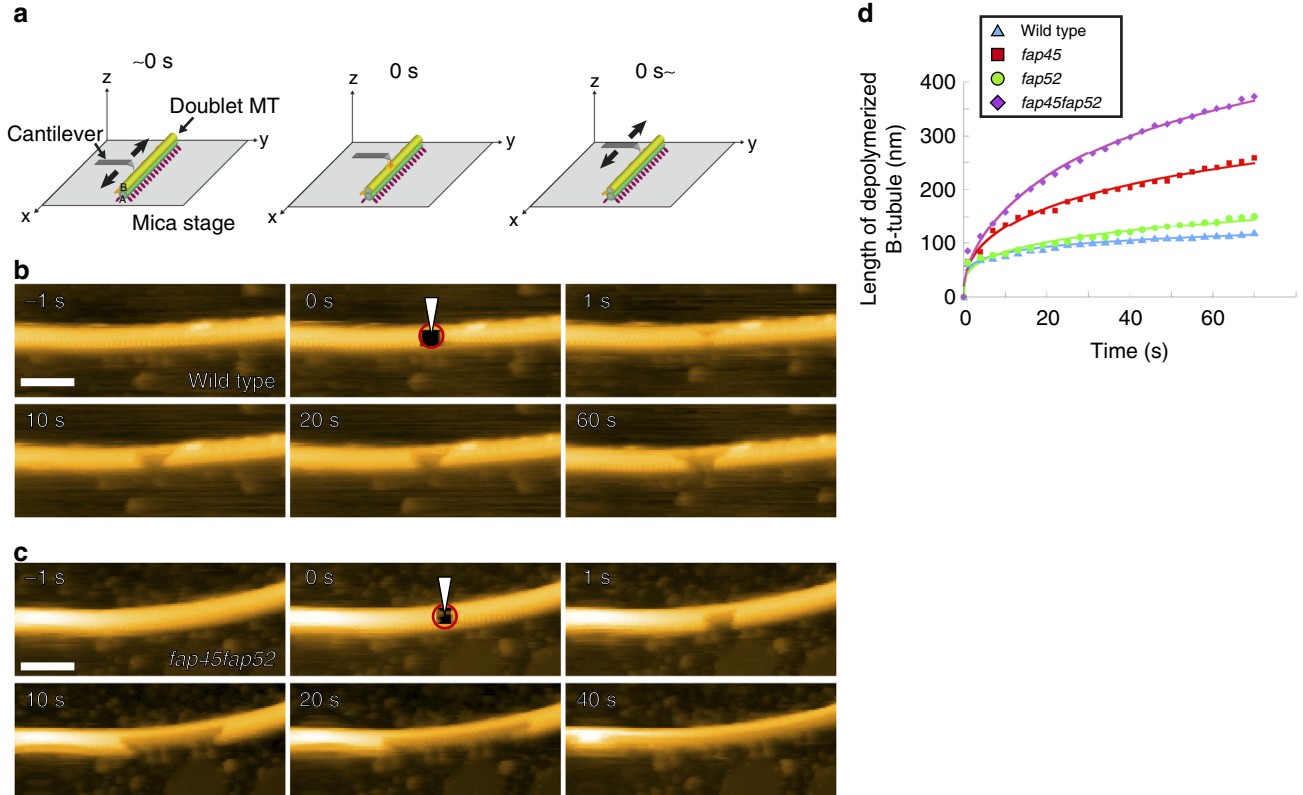

**Fig. 5** Direct observation of B-tubule depolymerization by HS-AFM. **a** Schematic of high speed-AFM observation of the DMT. The tip of the cantilever (tip diameter ~1–2 nm) intermittently tapped the DMT surface on a mica stage. At time 0 s, the tip of the cantilever was thrust into the B-tubule and made a small hole (circled in red in **b** and **c**), immediately followed by normal HS-AFM observation. **b**, **c** Representative time-lapse images of wild type (**b**) and *fap45fap52* (**c**) DMTs acquired by HS-AFM. In the wild type, enlargement of the hole usually stopped within a few seconds, whereas most of the B-tubules in the field were broken in *fap45fap52* by 40 s. Scale bar = 100 nm. **d** The average length of the depolymerized B-tubules was plotted against time (wild type: $n = 11$, total length = 7.25 μm; *fap45*: $n = 8$, total length = 4.92 μm; *fap52*: $n = 7$, total length = 3.83 μm; *fap45fap52*: $n = 10$, total length = 4.95 μm)

suggesting that both of the A and B-tubule were immobilized on the mica (Supplementary Fig. 7b); Class 3: The radial spokes are periodically and horizontally projected from the A-tubule, suggesting that the B-tubule was visualized on the top while the A-tubule was immobilized on the mica (Supplementary Fig. 7c).

To determine how a defect influences the stability of DMTs, we selected DMTs in the class 2 and 3 and made a small hole by thrusting the tip of the cantilever into the B-tubule (Time = 0, Fig. 5a) and monitored whether or not the damage propagates. In wild type, the hole rarely grew during the observation period (Fig. 5b, Supplementary Movie 5). On the other hand, in the mutants, the hole gradually expanded in both the proximal and distal directions (Fig. 5c, Supplementary Movie 6–8). We did not observe significant difference between depolymerization toward the plus end and the minus end of the DMTs. Time-course analyses of the depolymerized B-tubule length demonstrated that the tendency for depolymerization followed the order as *fap45fap52* > *fap45* > *fap52* ≥ wild type (Fig. 5d). These data strongly suggest that FAP45 and FAP52 stabilize the doublet B-tubule and prevent depolymerization induced by damage to the MT wall. Of note, the DMT of *fap45fap52* are still more stable than cytoplasmic microtubules, suggesting that B-tubules are also stabilized by other mechanisms, such as post-translational modifications of tubulin[37,38] (Supplementary Fig. 8) and fMIPs that bind inside the B-tubule along its length[35].

**MIP3a is important for the crossbridge of the inner junction.** Since MIP3a is located near the inner junction between the A-tubule and B-tubule and appeared to attach to the A-tubule[35], we

investigated whether FAP52 is involved in stabilizing the junction. We previously reported that FAP20 is a component of the inner junction and a null mutant of FAP20 (*fap20*) showed an abnormal swimming phenotype, although the structure of the DMTs appeared to be normal[39]. Based on the hypothesis that MIP3a anchors the B-tubule to the A-tubule together with FAP20, we constructed the double and triple mutants *fap20fap45*, *fap20fap52*, and *fap20fap45fap52*. Most *fap20fap45* cells were trembling and their motility appeared slightly worse than that of *fap20* cells (Supplementary Movie 9). In contrast, *fap20fap52* and *fap20fap45fap52* cells were essentially paralyzed (Supplementary Movie 9). Furthermore, the flagellar lengths of *fap20fap52* and *fap20fap45fap52* deviated greatly from wild type, *fap20*, and *fap20fap45* (Fig. 6a and b), with a far higher ratio of short flagella. To identify the cause of these defects, we observed the axonemes of the mutants by thin section TEM. The axonemes and DMTs of *fap20fap45* appeared similar to those of *fap20* (Fig. 6c and d). Interestingly, several doublet B-tubules in *fap20fap52* and *fap20fap45fap52* were detached from the A-tubules (Fig. 6c arrowheads; Fig. 6d) whereas no detachment of B-tubules from A-tubules was observed in *fap20fap45* axonemes. Given that such defects were rarely observed in *fap20* axonemes[39], these results demonstrate that MIP3a is required for anchoring the B-tubule to the A-tubule at the inner junction together with FAP20, and the junction is important for stabilizing the axonemal structure.

**Discussion**
Ciliary/flagellar MTs are remarkably stable compared with cytoplasmic MTs. Of the 9+2 MTs, DMTs are directly stressed by the

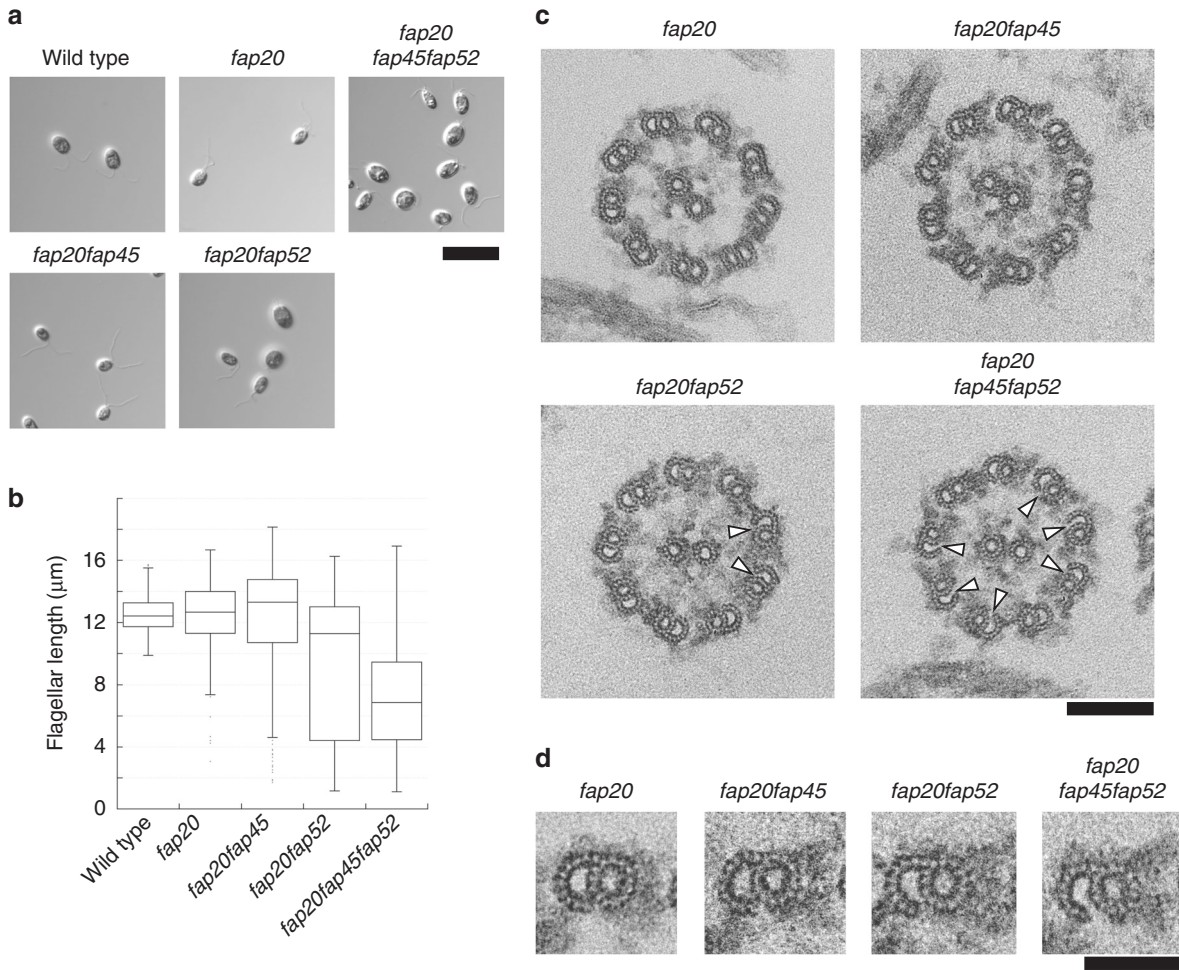

**Fig. 6** MIP3a is important for B-tubule anchoring to the A-tubule at inner junctions. **a** Differential interference contrast (DIC) images of wild type, *fap20*, *fap20fap45*, *fap20fap52*, and *fap20fap45fap52* cells. Scale bar = 10 μm. **b** Box and whisker plots of flagellar length in wild type, *fap20*, *fap20fap45*, and *fap20fap52*. *fap20fap52* and *fap20fap45fap52* flagella were significantly shorter than in wild type, *fap20*, and *fap20fap45*. The whiskers represent 95% confidence intervals. The box represents the interquartile range and the center line shows the median. **c** Cross-sectional EM images of *fap20*, *fap20fap45*, *fap20fap52*, and *fap20fap45fap52* axonemes. Arrowheads indicate disconnections between the A-tubule and B-tubule. Scale bar = 100 nm. **d** Cross-sectional images of *fap20*, *fap20fap45*, *fap20fap52*, and *fap20fap45fap52* DMTs. Scale bar = 50 nm

axonemal dynein-generated force and constantly bent and straightened in *Chlamydomonas* flagella or trachea cilia, yet their structures remain intact. The mechanism that stabilizes axonemal MTs has not been clarified. Here, we identified FAP45 and FAP52 as proteins that stabilize DMTs by binding to the inner lumen of the B-tubule. In this report we identify factors essential for the stabilization of ciliary/flagellar MTs.

Based on our studies and existing data, we propose a schematic model of FAP45 and FAP52 (Fig. 7b). Our cryo-ET observations show that MIP3a and MIP3c are composed of FAP52 and FAP45, respectively (Fig. 7a). FAP52/MIP3a anchors the protofilaments A13 and B10, whereas FAP45/MIP3c is bound inside of B7, B8, and B9 protofilaments. Besides, biochemical crosslinking shows that FAP45 and FAP52 directly interact with each other. Thus, these two proteins bundle B7–B10 protofilaments to reinforce the B-tubule. Consistent with this model, the B-tubules of *fap45* were more vulnerable to physical stress than those of *fap52* in the HS-AFM observation (Fig. 5d, Supplementary Movie 7). The MIP3a structure was observed also in vertebrates[7,24], but MIP3c was not clearly described probably due to the limitation of resolution. However, the amino acid sequence of FAP45 is highly conserved from protists to mammals (Supplementary Fig. 6), suggesting that

the functions of FAP45 are conserved also in higher organisms as in *Chlamydomonas*.

Our data also suggest that DMTs are stabilized by fail-safe mechanisms. The single mutants of FAP45 and FAP52 did not show significant decreases in swimming speed, and only the double mutant *fap45fap52* showed slow swimming and defects of the B-tubules. Although the B-tubules of *fap45fap52* were more easily depolymerized than wild type B-tubules, the speed of depolymerization was slower than that of in vitro polymerized pure MTs.

The inner junctions of DMTs are also stabilized by fail-safe mechanisms involving multiple proteins. We previously demonstrated that FAP20 is a constituent of the inner juction[39]. In addition to the true inner junction, our data clarified that MIP3a also connects B-tubules to A-tubules (Fig. 6c and d). Furthermore, some inner junctions appeared to be attached in *fap20fap52* and *fap20fap45fap52* mutants (Fig. 6c). In the triple mutant, additional proteins, such as tektin and PACRG, may contribute to stabilizing the inner junction, given our previous observation that these two proteins were partially retained in *fap20*[39].

Ciliopathy caused by FAP45/CCDC19 mutation has not been reported to date. On the other hand, patients lacking FAP52/

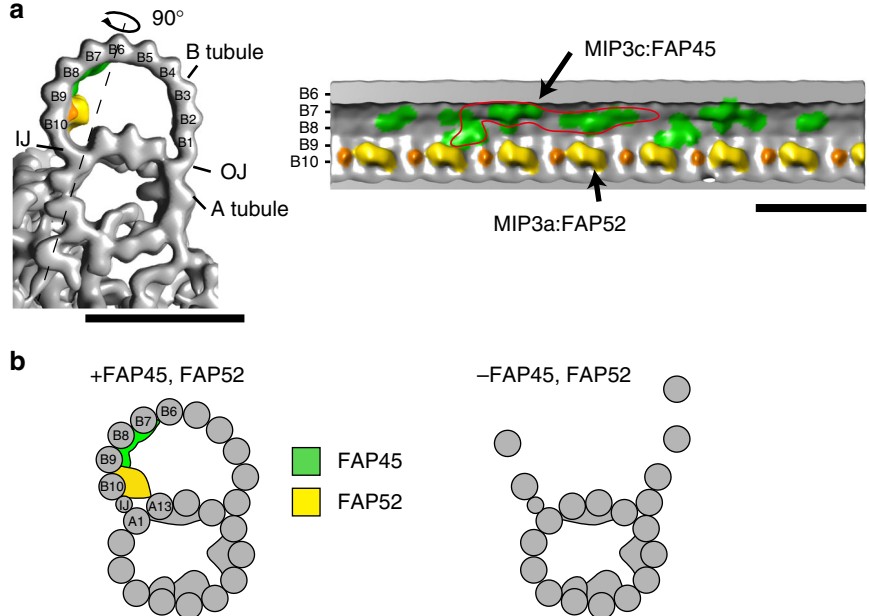

**Fig. 7** A schematic model of MIP3. **a** Isosurface rendering of the wild type density map. MIP3a-c is highlighted as shown in Fig. 2 (MIP3a: yellow; MIP3b: orange; MIP3c: green). The right map is the longitudinal view sliced by the line in the left map. A possible unit of MIP3c is circled in red. Scale bar = 25 nm. **b** A model of DMT stabilization. FAP45 (green) and FAP52 (yellow) are localized on the inner lumen of the B-tubule and stabilize the MT wall. In particular, FAP45 ties protofilaments B7–B9. FAP52 anchors B9 and B10 to A13 and binds to FAP45. Thus, these proteins strengthen the MT wall against DMT-bending. Loss of both proteins leads to destabilization of the B-tubule and the B-tubule is depolymerized by bending

WDR16 were reported to have abdominal situs inversus and situs inversus totalis[30]. Despite abnormal laterality, those patients did not have typical symptoms of primary ciliary dyskinesia, such as recurrent bronchiolitis, hydrocephalus, or low nasal nitric oxide levels. These findings are consistent with the *fap52* phenotype in *Chlamydomonas*, where we could not detect abnormal phenotypes other than the lack of MIP3a structure. Given that the phenotype of *fap45* was more severe than that of *fap52* in *Chlamydomonas*, we predict that the lack of FAP45/CCDC19 in human causes ciliopathy.

The structures of MIPs in the A-tubule are more complex than in the B-tubule. It is previously shown that hyper-stable ribbons were composed of four protofilaments to which MIP4 is bound[40], suggesting MIP4 stabilizes the ribbon structure. A recent paper reported that FAP85 is a MIP in the A-tubule and stabilizes cytoplasmic microtubules in vitro[22]. Besides, a study using *Tetrahymena* revealed that Rib72A and B are essential for assembly of MIP6, which is important for proper flagellar beating[23]. Although how those MIPs contribute to the stability of the A-tubule is still unclear, it is possible that MIPs in A-tubules also have fail-safe mechanisms, similar to MIP3a and c.

## Methods

**Strains and culture conditions**. The *Chlamydomonas* strains used in this study are listed in Supplementary Table 3. The double and triple mutants were constructed by standard methods[3]. All cells were cultured on Tris-acetate-phosphate (TAP) plates with 1.5% agar or in TAP medium. *fap45* and *fap52* mutants were isolated from a library of mutants generated by *aphVIII* gene insertion[31]. The mutants were backcrossed with wild type several times before use. For the PCR analyses, following primers were used; FAP45-F: 5′-GTGGCTTGAGCACCCTACTCACTT-3′; FAP45-R: 5′-CCGCTTGCTACCTCCAAATAAAGA-3′; FAP52-F: 5′-CGCCGAC CTCTACATTTCTGAGTT-3′; FAP52-R: 5′-CTTTGACTCCAGGTCCCAGATG AT-3′; *aphVIII*-F: 5′-GTCGACTTGGAGGATCTGGACGA-3′. For the probe of the southern blot, the *aphVIII* coding sequence was amplified with following primers; 5′-ATGGACGATGCGTTGCGTGC-3′; 5′-TCAGAAGAACTCGTCCAACA GCCG-3′. The probe was PCR-labeled with biotin-16-dCTP and detected by Streptavidin-HRP.

**Motility assay using the CLONA system**. To assay the motilities of *Chlamydomonas* cells in TAP medium or TAP with ficoll, videos were recorded using a high-speed camera (EX-F1; Casio) attached on a dark-field light microscope (BX51; Olympus) at 600 fps. The videos were analyzed using the CLONA system[41] and the results were plotted using R Project software.

**Antibodies**. The antibodies used in this study are described in Supplementary Table 4. Anti-FAP45 and anti-FAP52 antibodies were raised against full-length FAP45 and FAP52 proteins, respectively (Supplementary Fig. 2a). The full-length cDNA of FAP45 or FAP52 was cloned between the EcoRI and BamHI sites of pMAL-c2x (New England Biolabs). In both constructs, a 6×His tag was inserted into the HindIII site of pMAL-c2x to enable purification in two steps. For affinity purification, each cDNA was also cloned between the NdeI and EcoRI sites of pColdI (Takara). Expression of the recombinant proteins was induced in *Escherichia coli* BL21(DE3) (TAKARA BIO Inc.) with 0.3 mM IPTG, and almost all of the expressed protein was solubilized from each construct. MBP-FAP45-6×His and MBP-FAP52-6×His were purified with amylose resin (New England Biolabs) and then further purified with Ni-NTA agarose (Qiagen). These purified proteins were used as antigens. Each antibody was affinity purified using polyvinylidene difluoride membranes blotted with FAP45-6×His or FAP52-6×His. For western blot analyses, we loaded 5 μg of isolated axonemes in each lane. Full blot images are shown in Supplementary Fig. 9.

**Chemical crosslinking of axonemes**. Isolated axonemes of wild type, *fap45*, and *fap52* were treated with EDC (Thermo Scientific) in HMEK (30 mM Hepes–KOH, pH 7.4, 5 mM MgSO₄, 1 mM EGTA, 50 mM K-acetate) for 60 min at room temperature. After quenching the reaction, crosslinked axonemes were analyzed by SDS–PAGE and western blotting with anti-FAP45 and anti-FAP52 antibodies. The crosslinked products were immunoprecipitated with anti-FAP45 antibody following a previously described method[42]. Mass spectrometry analysis (LC tandem MS) of the precipitates was performed at the Proteomics and Mass Spectrometry Facility (University of Massachusetts Medical School).

**Immunofluorescence microscopy**. Nucleoflagellar apparatus (NFA) was prepared as previously described[43]. After fixation with 2% formaldehyde for 10 min at room temperature, NFAs were treated with cold methanol (−20 °C). The samples were immunostained as previously described[44]. Images were taken with a CCD camera (ORCA-R2; Hamamatsu Photonics) linked to a fluorescence microscope (IX70; Olympus).

**Generation of BCCP-tagged strains**. Fragments from the start codon to immediately before the stop codon in the FAP45 and FAP52 genes were amplified by genomic PCR and inserted into pIC2 plasmids[16]. In the FAP52 construct, a 3×HA tag was inserted into the C terminus of FAP52. Each construct was linearized and transformed into *fap45* or *fap52* cells by electroporation. Streptavidin-Alexa546 staining of the axonemes was performed as previously described[39].

**Thin-section TEM**. Samples for thin-section TEM were prepared as previously described[45] except that we used 50 mM Na-phosphate, pH 7.0 instead of cacodylate buffer. The samples were observed using a transmission electron microscope (JEM-3100FEF; JEOL) equipped with a 4096 × 4096-pixel CMOS camera (TemCam-F416; TVIPS). All images were taken at 300 keV, with ~3 μm defocus, at a magnification of 40,000 and a pixel size of 2.5 Å.

**Cryo-sample preparation**. Purified axonemes in HMDEK buffer (30 mM Hepes–KOH (pH7.4), 5 mM MgSO$_4$, 1 mM dithiothreitol, 1 mM EGTA, 50 mM potassium acetate, protease inhibitor cocktail (Nacalai Tesque)) were incubated with anti-beta-tubulin antibody (1:10,000 final; T0198, SIGMA) for 15 min on ice, followed by incubation with goat anti-mouse IgG (H+L) 15 nm gold (1:20 final; BB International) and 15 nm colloidal gold conjugated with BSA. The mixtures were loaded onto home-made holey carbon grids and plunged into liquid ethane at −180 °C using an automated plunge-freezing device (EM GP; Leica).

**Image acquisition**. Grids were transferred into the JEM-3100FEF using a high-tilt liquid nitrogen cryo-transfer holder (914; Gatan Inc.). Tilt series images were recorded at −180 °C using a K2 summit direct detector (Gatan) and the serialEM[46]. The angular range was ±60° with 2.0° increments. The total electron dose was limited to 100 e$^−$/Å$^2$ and the nominal magnification was ×6000. An in-column energy filter was used with a slit width of 20 eV and a pixel size of 7.1 Å.

**Image processing**. Image processing for subtomogram averaging of DMT was carried out as previously described[33,34]. Tilt series images were aligned and back-projected to reconstruct 3D tomograms using IMOD[47]. Alignment and averaging of the subtomograms were performed using custom Ruby-Helix scripts[48] and PEET software[7] to average the 96-nm repeats of DMTs. UCSF Chimera was used for isosurface renderings[49]. FSC curves were shown in Supplementary Fig. 5f.

**HS-AFM observation**. Flagella were demembranated with 0.5% Nonidet P-40 in HMDEK (30 mM Hepes–KOH, 30 mM Hepes–KOH, pH 7.4, 5 mM MgSO$_4$, 1 mM DTT, 1 mM EGTA, 50 mM K-acetate, protease inhibitor cocktail (Nacalai Tesque)), followed by centrifugation for 2 min. The axoneme pellets were suspended with ATP buffer (HMDEK containing 0.1 mM ATP, 1 mM ADP, and 0.5% polyethylene glycol (20,000) and incubated for 1 min at room temperature. Frayed axonemes were then diluted with HMDEK and placed on a mica stage. After 10 min incubation, residuals are washed with the HMDEK buffer. The AFM images were recorded using a home-built high-speed atomic force microscope based on tapping mode[50–52] at frame rates of 1–2 fps. All observations were performed in HMDEK buffer at room temperature. We selected DMTs of which the angle to the fast-scan axis is <45° to minimize accidental damage on the DMTs. A miniature AFM cantilever (Olympus) with a spring constant of ~0.2 N/m, a quality factor of ~2, and a resonant frequency of ~800 kHz was used for the HS-AFM imaging. For imaging DMTs, a free oscillation amplitude was set to be ~2 nm and the set-point amplitude for the feedback control was 80% of the free oscillation amplitude, corresponding to the tapping force of ~120 pN[53] (used the Eq. (10)). To create defects on the DMTs, the set-point amplitude was decreased to <5% of the free oscillation amplitude and the tapping force was increased to >200 pN at the specified area.

## Data availability

Data supporting the findings of this manuscript are available from the corresponding author upon reasonable request. A reporting summary for this article is available as a Supplementary Information file. The source data underlying Figs. 1e–g, 5d and 6b are provided as a Source Data file. Cryo-ET reconstructions are available from the Electron Microscopy Data Bank (EMDB) with following accession numbers: fap45 (EMD-9766), fap52 (EMD-9767), and wild type (EMD-9768).

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

## Acknowledgements

We thank Dr. John Leszyk (UMMS) for mass spec analysis and Dr. Brian Dynlacht (NYU) for helpful discussion. M.O. is a recipient of the fellowship from Japan Society for Promotion of Sciences for young scientists. This work was supported by CREST, the Japan Science and Technology Agency to M.K. and T.A., the Japan Society for the Promotion of Science (JSPS) KAKENHI (18H01837) to T.U., JSPS KAKENHI (25117506 and 15H01206) to K.W., and JSPS KAKENHI (16H04805) and JST Advanced Low Carbon Technology Research and Development Program (ALCA, JPMJAL1105) to H.F.

## Author contributions

M.O. and M.K. designed the experiments. M.O. performed most of the experiments and analyzed the data. M.O., T.U. and M.K. wrote the paper. M.O. and K.W. performed chemical crosslinking experiments. M.O. and T.U. performed A.F.M. analyses. M.O. and H.Y. performed cryo-ET and E.M. experiments. T.Y., H.I., and H.F. provided the *Chlamydomonas* mutant library. T.A. devised the AFM method for sample manipulation.

## Additional information

**Competing interests:** The authors declare no competing interests.

