## [Peer Review File · Nature Communications]

Reviewers' Comments:

Reviewer #1:

Remarks to the Author:

This paper from the Kikkawa group is identifying the microtubule inner proteins inside the B-tubule of the microtubule doublet, namely FAP45 and FAP52. In this paper, the group identifies the two proteins using the mass spectrometry, pinpoint the location of the two proteins using cryo-electron tomography, identify the functions of the proteins as stabilizing the B-tubule by looking at microtubule depolymerization using double and triple mutants using atomic force microscopy.

The paper is well-written so that it is very easy to read. The paper is really solid in terms of the experiment carried out to identify and characterize the MIPs. This is the first paper where MIPs in the B-tubule has been identified and their functions are clearly illustrated with the elegant atomic force microscopy experiment. Also, the FAP45 is probably the first filamentous MIP identified following the first report of it (Ichikawa et al., 2017, Nat Comm).

Just to note that there is a paper where Rib72a and Rib72b were identified as MIPs in the A-tubule (<https://www.molbiolcell.org/doi/pdf/10.1091/mbc.E18-06-0405>). So this paper is important to the field, now having to identify MIPs in both A and B-tubules and understand their functions. Also, the authors should cite the new paper from the Nicastro group.

Here are my comments:

- With the EDC crosslinking, what is the identity of proteins from other high molecular weight bands?
- Since FAP52 is conserved in many organisms, including Tetrahymena, why don't the authors dock the FAP52 in the map from Tetra at subnanometer resolution (Ichikawa et al., 2017). That will prove the point that the structure of FAP52 consists of two beta propellers and fit well into MIP3a.
- The authors named FAP45 with the name MIP3c but this is already named in Ichikawa et al., 2017 as fMIP B8B9 and again FAP45 is conserved in Tetrahymena. The authors should point to this fact from that paper.

Here are my minor comments:

- The last sentence in Introduction: microtubule inner protein should be abbreviated as MIP
- Page 11: Since MIP3a is located near the inner junction between the A- and B-tubules and appeared to attach to the A-tubule 28. The citation should be 29

Reviewer #2:

Remarks to the Author:

The manuscript entitled "Inner lumen proteins stabilize doublet microtubules in cilia/flagella" describes the localization and role of two as-yet uncharacterized, evolutionarily conserved ciliary/flagellar proteins, FAP45 and FAP52. The cryo-ET analyses of the *Chlamydomonas* mutants lacking the above-mentioned proteins led Authors to the conclusion that FAP45 and FAP52 are B-tubule luminal proteins, MIP3c and MIP3a, respectively.

The presence of the luminal proteins in both, A- and B-tubules were earlier described (Nicastro et al., 2006, Science, 2011; Pigino et al., 2012, J. Struct. Biol.; Sui and Downing, 2006, Nature; Maheshwari et al., 2015, Structure; Ichikawa et al., 2017, Nat Commun) however, the identity of the proteins building these densities and their role remained elusive. Recently, it was reported that FAP85 is likely an A-tubule MIP (Kirima and Oiwa, 2017, Cell Struct Funct) while Rib72A and Rib72b are required for the assembly of the A-tubule MIP1, 4, and 6 (Stoddard et al., 2018, Mol Biol Cell).

The evidence suggesting that FAP45 and FAP52 are two B-tubule MIPs can be an important piece of data. The functional studies of the *fap45* and *fap52* (single, double and triple – with *fap20*) mutants could shed a light on the molecular basis of cilia stability and resistance of cilia/ flagella to forces generated during their bending/beating.

In my opinion, these data will be of interest to the scientific community. However, some additional work is needed.

I have found the following issues that should be addressed by the Authors (I apologize if I misunderstood the Authors):

General remarks:

The writing of the manuscript could be improved (please see minor remarks listed below). I would suggest presenting the structure of the *fap45* and *fap52* 96-nm repeat as a part of the main figure, point to the differences in the dynein b density (WT versus mutants) and next describe lack of MIPs in B-tubule.

Authors should also provide a better characterization of the analyzed *Chlamydomonas* mutant cells (in *Chlamydomonas* constructs are inserted randomly into the genome – non-homologous recombination): please, show PCR analyses of the targeted loci, and Southern blot to ensure that only one locus was targeted in a single mutant (see below). A phenotype of a single mutant is very similar to the one observed in the wild type (*fap52*) or is manifested by slightly slower swimming rate (*fap45*). Thus, the rescue experiment may not be sufficient (the phenotype of the rescued clones was not documented by the Authors).

Besides the lack of MIPs (MIP3c or MIP3a) in B-tubule, both mutants (*fa45* and *fap52*) have reduced dynein b density. Although unlikely, it is possible that genes encoding dynein b or its tail are affected.

It would be also of interest to analyze the localization of BCCP tagged *fap45* and *fap52* (this would support the cryo-ET analyses of the mutants).

Minor remarks

Abstract:

“To address this question, we identified a new class of microtubule-associated proteins, named FAP45 and FAP52...”

MIPs are a new class of proteins, here is described the identification of two of MIPs

Introduction:

“The structures and related genes of cilia and flagella are well conserved among eukaryotes. Cilia are classified as either non-motile or motile.....”

I would suggest to include the sentence “The structures and related genes of cilia and flagella are well conserved among eukaryotes” in the second paragraph. Such sentence in the paragraph describing also primary cilia implies that also primary cilia have a structure (and protein composition) similar to flagella. I would also add “motile” before “cilia” and instead of “related gene” used ciliome or protein composition.

“The activity of dyneins is regulated by the interaction between the radial spokes and the central pair of singlet MTs10-12.”

Please mention a role of the N-DRC and cite the appropriate papers.

"Recent advances in cryo-ET techniques have dramatically revealed structural details... Please replace the word "dramatically" or re-write the whole sentence, cite references

"Nicastro and colleagues have provided insights into the structural basis of DMTs using cryo-ET of *Chlamydomonas* flagella. They reported periodic high densities on the inner surfaces of A-tubules and B-tubules, which they named microtubule inner proteins (MIPs, Fig.1a)^{7, 17}. MIPs have also been observed in the axonemes of higher organisms^{7, 17-20}, implying.."

Please cite also the work of other groups (see above)

Results:

Identification of proteins essential for DMT stabilization in flagella

"To identify proteins that stabilize DMTs, we searched the *Chlamydomonas* flagellar proteome database²¹ using the assumption that...

Please re-phrase "using the assumption"

"However, the functions of FAP45/CCDC19 are totally unclear and thus we focused on FAP45."

Please re-phrase

"We first investigated the partner that interacts..."

Looked for, attempted to identify...partner(s)

"In addition, a mass spectrometric analysis revealed that the ~130 kDa product is composed of FAP45 and FAP52 proteins, probably in a 1:1 ratio.."

What is the molecular mass of FAP52? Please, provide first information about MW of FAP52. (before MW of the complex)

How was calculated the ratio of FAP45:FAP52?

Fig. 1b - Why there is an increase of the ODA IC2 signal in *fap45* mutant and *fap45fap52* double mutant compared to WT and *fap52* mutant?

Fig. 1c, d – please add loading control.

Isolation and characterization of *Chlamydomonas* FAP45 and FAP52 mutants

Fig. S1 - FAP45 and FAP52 mutants analyses – please provide PCR analyses of the targeted loci and Southern blot analyses of the genomic DNA of WT and mutant cells to ensure that a single locus was targeted in these *Chlamydomonas* mutant (according to Gonzalez-Ballester et al., 2011, about 90% of the transformants harbored a single copy of the integrated marker DNA, but about 10% may have two copies (compare with supplement figures Yamano et al., 2015, PNAS, Characterization of cooperative bicarbonate uptake into chloroplast stroma in the green alga *Chlamydomonas reinhardtii*).

"On the other hand, *fap45* and *fap52* axonemes retained the known major axonemal components, such as outer arm dyneins, inner arm dyneins, radial spokes, and the dynein regulatory complex (N-DRC) (Fig. 1b)."

Fig. 1b – western blot is showing only level of some proteins, it would be more convincing if you could provide an image of the structure of the 96-nm unit of fap45 and fap52 mutant obtained using cryo-ET (main figure)

“Next, we examined the motility phenotype of fap45 and fap52. The swimming velocity and beat frequency of fap45 cells were slightly reduced” (Fig. 1e, Movie 1),

Please record movies showing how the flagella beat.

“Since the medium under confluent culture conditions is more viscous than that of log phase conditions,...”

Please cite references

“Most of the wild type cells swam slowly but smoothly (Movie 3) whereas many of the fap45fap52 cells stopped swimming or struggled to swim against the viscous load...”

Please provide the number of cells not swimming / slowly swimming cells. Can you record and measure the cells trajectories?

“Since the fap45fap52 axonemes had normal levels of axonemal dyneins, radial spokes, and N-DRC, these results suggest that the lack of both FAP45 and FAP52 causes structural defects in the DMT.”

1. FAP45 and FAP52 may form a minor complex that either links the major complexes or regulates their function.
2. Dynein b is reduced in mutants
3. Information about the normal level of axonemal major complexes is repeated a 3rd time.

FAP45 and FAP52 are luminal proteins in B-tubules

“...half of the FAP45 was extracted at 0.3%, and the protein was completely extracted from the pellet at 0.7% (Supplementary Fig. S2b).”

“These axonemes were purified from rescue strains expressing FAP proteins whose N- or C-terminus was fused to biotin carboxyl carrier protein tag (BCCP tag, Fig. S2b and c)..”

Inconsistency in Fig S2

Fig. S2b- WB of the sarkosyl-treated axonemes

Fig. S2c – WB of FAP45-BCCP, WB of FAP52-BCCP

Fig. 2Sb – letter “B” should be moved up to the upper corner (is in the lower left corner (similar in Fig2c), in case of Fig S2A and FigS2d, letters “a” and “d, e” are in the upper left corner.

Lack of FAP45 and/or FAP52 causes structural defects in B-tubules

“In both our Chlamydomonas DMT and Tetrahymena DMT28,..”
Is the REF correct?

Fig.2 – e, and f (change fap45+ MIP3c to fap45=MIP3c), similar for fap52. “+” is misleading.

“All of the protofilaments in the B-tubules were completely missing in some DMTs, whereas DMTs remaining several protofilaments in the B-tubules were also observed.”

Please provide numbers (%).

"...we also found that the density of dynein b, a species of inner arm dyneins was decreased in fap45 and fap52 (Supplementary Fig. S3a-c)."

Earlier it was stated that mutants had a normal level of axonemal dyneins. I would suggest showing images of 96-nm repeats as in the main figure, describe the reduced level of dynein b and next changes in B-tubule luminal proteins.

Additionally, localization of the BCCP tagged FAP45 and FAP52 can be analyzed by cryo-ET.

MIP3a is important for B-tubule anchoring to the A-tubule at the inner junction of DMT

Discussion

Data presented in FigS5 should be described in the Results section

"This suggests that B-tubules are also stabilized by other mechanisms, such as tubulin acetylation 32"

Please cite other REF

Please, discuss data presented by:

Kirima and Oiwa, 2017, Cell Struct Funct

Stoddard et al., 2018, Mol Biol Cell

Methods

Please provide:

1. Silver-stained gel and western blot of the purified proteins used to produce antibodies.
2. Silver-stained gel of Fap45 and Fap 52 precipitates

How many ug of the proteins was loaded on the gel (western bots)

Table S1 is showing only selected proteins and thus, it is difficult to make one's own mind about these results. Can you include also proteins identified by the lower number of peptides?

Table S2 is showing peptides of only two proteins. Please show all proteins (including their MW) identified by mass spectrometry. Was corresponding fragment of the gel (control) analyzed by mass spec? Any proteins in the control?

Reviewer #3:

Remarks to the Author:

The paper "Inner lumen proteins stabilize doublet microtubules in cilia/flagella" by Owa et al. uses electron tomography, high-speed atomic force microscopy and cellular imaging as functional assays to investigate the structural and stabilization role of FAP45, FAP52 and FAP20 in microtubuli.

The work identifies the location of these factors in the tubules (based on tomographic maps) and provides data that indicate that these proteins are essential for native flagella function, especially when exposed to physical resistance (functional imaging), and that the proteins convey structural stability to the B-tubule in the assembly (based on invasive HS-AFM imaging).

The paper is well written and provides novel insights. The data is of highest quality.

My major point concerns the HS-AFM imaging part. In this part of the manuscript, the authors apply through precise force application a small defect in the B-tubule and then monitor the progression of this defect (depolymerization of the tubule) as a function of time.

Major concern 1:

The experiment relies on the concept that the doublet microtubule is attached with a very precise geometric constraint to the surface with the A-tubule immobilized and the B-tubule standing right up facing the HS-AFM tip. This is of such crucial importance for the entire conclusions that it must be precisely described and also conclusively documented in the main text.

Major concern 2:

The progression of the defect is a qualitative measure. It will in HS-AFM depend on many parameters, such as tip geometry, scan direction and force.

2.1 Comment on the tips used and whether the experiments are performed with the same or essentially the same tip, and how this is controlled.

2.2 The authors mainly (only) present results where the MT is oriented in line with the fast scan axis (horizontal in the image). Were the experiments also performed on differently oriented MTs, was the depolymerization then different, what were accepted angular deviations from the horizontal axis, etc?

2.3 The scanning force is obviously the most important parameter, as the depolymerization is induced by a force induced defect it is likely dependent on force. The applied force in a HS-AFM depends on the free oscillation amplitude $A(\text{free})$ and the setpoint amplitude $A(\text{set})$. The authors do not comment in detail on this. It has recently been shown (Miyagi, RSI, 2016) that both $A(\text{free})$ and $A(\text{set})$ can and must be controlled to control the applied tip-surface interaction. In the same work it has been shown that this is crucial for the lifetime of weak intermolecular interactions. Thus, the author must detail $A(\text{free})$ and $A(\text{set})$ and how $A(\text{free})$ was stabilized over extended experimental periods. Obviously, $A(\text{free})$ and $A(\text{set})$ must be quasi identical in all experiments (WT, fap45, fap52, fap45fap52) to be compared.

Minor concerns:

1 The false color scale for the representation of the HS-AFM data is an unusual and inappropriate choice (Figure 4). While the supplementary movies are very clear, the figure displays the highest protruding structures in a teal tone that is not intuitively associated with stronger protrusion than the green and yellow. It almost looks like a contrast inversion and when printed in gray scales all these colors have the same intensity.

2 Plotting the progression of depolymerization towards proximal and distal directions would be nice. Why is there no difference?

Abstract:

The mention of FAP20 occurs once, and it is unclear how it relates to the other findings.

Introduction:

The sentence "recent studies revealed that the assembly of cilia is significantly decreased in several types of tumors^{5,6}, suggesting a close relationship between cilia and tumorigenesis" seems like an overstatement. There are many things different in tumors, and thus "close relationship" sounds like a strong call.

Figure1:

Coloring the different elements in Figure1a would be beneficial for the understanding.

Introduction:

"Recent advances in cryo-ET techniques have dramatically revealed structural details" - delete "dramatically".

Results:

"To reveal the defect caused by the loss of FAP45 and FAP52, we observed the mutant axonemes by cryo-ET. Interestingly, the mutants had structural defects inside of the B-tubule (Fig. 2)." - "In agreement" rather than "Interestingly".

Reviewers' comments and replies for them:

Reviewer 1

- With the EDC crosslinking, what is the identity of proteins from other high molecular weight bands?

Although western blot analyses in Fig. 1c and d showed the high molecular bands, those were not as strong and discrete as FAP45-FAP52 band or FAP45-tubulin band in the silver staining of FAP45-IP products (Supplementary Figure S1d). Therefore, we did not analyze them by the mass spec.

- Since FAP52 is conserved in many organisms, including *Tetrahymena*, why don't the authors dock the FAP52 in the map from Tetra at subnanometer resolution (Ichikawa et al., 2017). That will prove the point that the structure of FAP52 consists of two beta propellers and fit well into MIP3a.

As suggested by the reviewer, we fit a WD40 dimer into the map (Ichikawa et al., 2017) and showed the image in Supplementary Figure S5e. We mentioned it in the paragraph; "Indeed, a model comprising two β -propellers fit well into the *Chlamydomonas* density map (Supplementary Fig. S5d) as well as the recently published sub-nanometer map of *Tetrahymena*³⁵ (Supplementary Fig. S5e), suggesting that the density of MIP3a corresponds to one molecule of FAP52." (page 9-10)

- The authors named FAP45 with the name MIP3c but this is already named in Ichikawa et al., 2017 as fMIP B8B9 and again FAP45 is conserved in *Tetrahymena*. The authors should point to this fact from that paper.

We apologize that we didn't quote the name, fMIP B8B9. We rewrote the paragraph and described fMIP B8B9; "These densities are arranged in a 48-nm repeat, and reminiscent of filamentous densities called fMIP B8B9 in *Tetrahymena* DMTs³⁵. MIP3 and fMIP B8B9 appear to be connected by a lateral filament, which is also missing in *gap45* mutant. Therefore, we renamed the missing structure as "MIP3c" (Fig 3e right and Fig. 7a)." (page 9)
In our results, the lateral density connecting MIP3 and fMIP B8B9 is also missing in *gap45*. Therefore, we renamed those densities as MIP3c.

- The last sentence in Introduction: microtubule inner protein should be abbreviated as MIP

We corrected it.

- Page 11: Since MIP3a is located near the inner junction between the A- and B-tubules and appeared to attach to the A-tubule 28. The citation should be 29

We corrected the citation.

Reviewer 2

*The writing of the manuscript could be improved (please see minor remarks listed below). I would suggest presenting the structure of the *gap45* and *gap52* 96-nm repeat as a part of the main figure, point to the differences in the dynein b density (WT versus mutants) and next describe lack of MIPs in B-tubule.*

We followed the suggestion. In the latest Fig. 2, we showed the 96-nm repeats of wild type and the mutants, and rewrote the paragraph. First we described the decreased density of dynein b,

followed by the inner structures of the B-tubule; "To reveal the defect caused by the loss of FAP45 and FAP52, we observed the mutant axonemes by cryo-ET. We found that the density of dynein b, a species of inner arm dyneins was decreased in *fap45* and *fap52* (Fig. 2a-c). This result suggests that the number of dynein b may be reduced or binding of dynein b to the B-tubule in the mutants may be more flexible than in wild type.

For further visualization of missing structures in *fap45* and *fap52*, we applied Student's t-test" (page 8-9)

Authors should also provide a better characterization of the analyzed Chlamydomonas mutant cells (in Chlamydomonas constructs are inserted randomly into the genome – non-homologous recombination): please, show PCR analyses of the targeted loci, and Southern blot to ensure that only one locus was targeted in a single mutant (see below). A phenotype of a single mutant is very similar to the one observed in the wild type (fap52) or is manifested by slightly slower swimming rate (fap45). Thus, the rescue experiment may not be sufficient (the phenotype of the rescued clones was not documented by the Authors).

We appreciate the important suggestion by the reviewer.

First, We added the description of the phenotypes of rescue strains in the text (page 8). Although those cells swam normally like wild type, phenotypes of the original mutants were really subtle and the rescue experiments were not enough to conclude that each of the mutants was single mutated.

Therefore, to show *aph-VIII* was only inserted into one locus, we performed a southern blot using the *aph-VIII* coding sequence as a probe. In both mutants, we detected a single band (Supplementary Figure S1c). Also, we showed PCR analyses around the insertion in Supplementary Figure S1b. Thus, we concluded that both *fap45* and *fap52* are single mutants. We mentioned that in the main text; " By a southern blot, we confirmed that each mutant has one *aphVIII* fragment only at the targeted locus (Supplementary Figure S1c)." (page 6)

It would be also of interest to analyze the localization of BCCP tagged *fap45* and *fap52* (this would support the cryo-ET analyses of the mutants).

We tried to amplify the BCCP density using the method we previously established (Oda et al., 2013 etc.). However, we had difficulty to visualize the tag due to low accessibility of streptavidin and cytochrome c to the inside of the B-tubule, as shown in Supplementary Figure S4.

"To address this question, we identified a new class of microtubule-associated proteins, named FAP45 and FAP52..."

MIPs are a new class of proteins, here is described the identification of two of MIPs

We rewrote the sentence as follows; "To address this question, we identified two uncharacterized proteins, FAP45 and FAP52, as microtubule inner proteins (MIPs) in *Chlamydomonas*." (page 2)

"The structures and related genes of cilia and flagella are well conserved among eukaryotes. Cilia are classified as either non-motile or motile....."

I would suggest to include the sentence "The structures and related genes of cilia and flagella are well conserved among eukaryotes" in the second paragraph. Such sentence in the paragraph describing also primary cilia implies that also primary cilia have a structure (and protein composition) similar to flagella. I would also add "motile" before "cilia" and instead of "related gene" used ciliome or protein composition.

We transferred the sentence to the beginning of the second paragraph as follows; "The structure and protein composition of motile cilia and flagella are well conserved among eukaryotes. The axoneme, the core structure of cilia and flagella, is composed of a central pair of microtubules..." (page 3)

“The activity of dyneins is regulated by the interaction between the radial spokes and the central pair of singlet MTs10-12.”

Please mention a role of the N-DRC and cite the appropriate papers.

We added a sentence on the N-DRC and cited papers as follows; " The N-DRC forms a cross-bridge between neighboring DMTs and is required for orchestrating dynein activity and axonemal integrity¹³⁻¹⁶." (page3)

13. Heuser et al., 2009 *The Journal of cell biology* 187, 921-933; 14. Lin et al., 2011 *Journal of Biological Chemistry*, jbc. M111. 241760; 15. Bower et al., 2013 *Molecular biology of the cell* 24, 1134-1152; 16. Oda et al., 2015 *Molecular biology of the cell* 26, 294-304.

“Recent advances in cryo-ET techniques have dramatically revealed structural details...

Please replace the word “dramatically” or re-write the whole sentence, cite references

We deleted "dramatically".

“Nicastro and colleagues have provided insights into the structural basis of DMTs using cryo-ET of Chlamydomonas flagella. They reported periodic high densities on the inner surfaces of A-tubules and B-tubules, which they named microtubule inner proteins (MIPs, Fig.1a)⁷.¹⁷. MIPs have also been observed in the axonemes of higher organisms^{7, 17-20}, implying..”

Please cite also the work of other groups (see above)

We cited the two papers; Kirima and Oiwa, 2017, *Cell Struct Funct*; Stoddard et al., 2018, *Mol Biol Cell*.

“To identify proteins that stabilize DMTs, we searched the Chlamydomonas flagellar proteome database²¹ using the assumption that...

Please re-phrase “using the assumption”

We re-phrased; "To identify proteins that stabilize DMTs, we searched the *Chlamydomonas* flagellar proteome database²⁷. We postulated that those proteins are (1)..." (page 5)

“However, the functions of FAP45/CCDC19 are totally unclear and thus we focused on FAP45.”

Please re-phrase

We re-phrased as follows; "However, the functions of FAP45/CCDC19 are unclear. Thus, we chose FAP45 as a candidate of proteins stabilizing DMTs." (page 5)

“We first investigated the partner that interacts...”

Looked for, attempted to identify...partner(s)

We re-phrased; "We first looked for partner(s)..." (page 5)

“In addition, a mass spectrometric analysis revealed that the ~130 kDa product is composed of FAP45 and FAP52 proteins, probably in a 1:1 ratio..”

What is the molecular mass of FAP52? Please, provide first information about MW of FAP52. (before MW of the complex)

How was calculated the ratio of FAP45:FAP52?

We added the information about MW of each protein in the sentence. We estimated the ratio based on the sum of MWs; " In addition, a mass spectrometric analysis revealed that the ~130 kDa product is composed of FAP45 (~59 kDa) and FAP52 (~66 kDa), probably in a 1:1 ratio based on the molecular weight (Fig. 1c and d, filled arrowhead; Supplementary Fig. S1d,

arrowhead; Supplementary Table S2)." (page 5-6)

*Fig. 1b - Why there is an increase of the ODA IC2 signal in *fap45* mutant and *fap45fap52* double mutant compared to WT and *fap52* mutant?*

As the reviewer pointed out, the intensity of ODA IC2 in *fap45* background axonemes seemed slightly higher than in WT and *fap52*. However, ODA is properly assembled and there is no extra density around ODA in *fap45* DMTs (by tomography). Therefore, we concluded that the difference of the intensity is not significant.

Fig. 1c, d – please add loading control.

Fig. 1c and d showed loading control in themselves. Please see un-crosslinked FAP45 (~59 kD in Fig. 1c) and FAP52 (~66 kD in Fig. 1d) bands. Since the protein level of FAP45 in *fap52* or FAP52 in *fap45* is almost same to that in wild type (Fig. 1b), the intensity of un-crosslinked FAP45 or FAP52 represents the loading amount of wild type and mutants under the same crosslinking condition.

*“On the other hand, *fap45* and *fap52* axonemes retained the known major axonemal components, such as outer arm dyneins, inner arm dyneins, radial spokes, and the dynein regulatory complex (N-DRC) (Fig. 1b).”*

*Fig. 1b – western blot is showing only level of some proteins, it would be more convincing if you could provide an image of the structure of the 96-nm unit of *fap45* and *fap52* mutant obtained using cryo-ET (main figure)*

We deleted the statement " such as outer arm dyneins, inner arm dyneins, radial spokes, and the dynein regulatory complex (N-DRC)" from the sentence. As the reviewer suggested, in addition to Fig. 1b, which shows the level of axonemal components, we add Figure 2, where cryo-ET data of the 96-nm repeat of the mutants are shown. Also, in the following section, we referred to the reduction of the dynein b density (page 8).

*“Next, we examined the motility phenotype of *fap45* and *fap52*. The swimming velocity and beat frequency of *fap45* cells were slightly reduced” (Fig. 1e, Movie 1), Please record movies showing how the flagella beat.*

We recorded high-speed movies (Movie 2) and add the description; "Flagellar beating of swimming cells in these mutants appeared normal compared to wild type (Movie 2)." (page 7)

“Since the medium under confluent culture conditions is more viscous than that of log phase conditions,…”

Please cite references

We did not find any references on the viscosity of confluent *Chlamydomonas* cultures and had difficulty to measure viscosity of *Chlamydomonas* cultures, so we deleted the statement and simply described as follows; "We also observed the swimming behavior of wild type and *fap45fap52* log phase cells in the viscous medium." (page 7)

*“Most of the wild type cells swam slowly but smoothly (Movie 3) whereas many of the *fap45fap52* cells stopped swimming or struggled to swim against the viscous load…”*

Please provide the number of cells not swimming / slowly swimming cells. Can you record and measure the cells trajectories?

67 of 78 cells didn't swim in viscous media and we added the information in the main text. We showed trajectories of swimming wild type cells and *fap45fap52* cells in Supplementary Figure S3.

*“Since the *fap45fap52* axonemes had normal levels of axonemal dyneins, radial spokes, and N-DRC, these results suggest that the lack of both FAP45 and FAP52 causes structural defects in the DMT.”*

1. FAP45 and FAP52 may form a minor complex that either links the major complexes or regulates their function.

2. Dynein b is reduced in mutants

3. Information about the normal level of axonemal major complexes is repeated a 3rd time.

As the reviewer suggested, still there are several possibilities for functions of FAP45 and FAP52 at this point. Therefore, we rewrote the sentence; "These results indicate that the lack of both FAP45 and FAP52 led to synergistic effects on motility phenotypes." (page 7)

“...half of the FAP45 was extracted at 0.3%, and the protein was completely extracted from the pellet at 0.7% (Supplementary Fig. S2b).”

“These axonemes were purified from rescue strains expressing FAP proteins whose N- or C-terminus was fused to biotin carboxyl carrier protein tag (BCCP tag, Fig. S2b and c).”

Inconsistency in Fig S2

Fig. S2b- WB of the sarkosyl-treated axonemes

Fig. S2c – WB of FAP45-BCCP, WB of FAP52-BCCP

Fig. 2Sb – letter “B” should be moved up to the upper corner (is in the lower left corner (similar in Fig2c), in case of Fig S2A and FigS2d, letters “a” and “d, e” are in the upper left corner.

We added new Supplementary Figure S2 and S3 so now we relabeled previous Fig. S2 as Fig. S4. We apologize that the figure number in the text was wrong. We corrected it; “...half of the FAP45 was extracted at 0.3%, and the protein was completely extracted from the pellet at 0.7% (Supplementary Fig. S4a).” (page 7)

Also, we apologize that the labeling in the Figure was confusing. We re-arranged some of the pictures and the labeling.

*“In both our *Chlamydomonas* DMT and *Tetrahymena* DMT28,..”
Is the REF correct?*

We corrected it to Ichikawa et al., 2017.

*Fig.2 – e, and f (change *fap45+* MIP3c to *fap45=MIP3c*), similar for *fap52*. “+” is misleading.*

We relabeled them as "MIP3c (t-map) on *fap45*" and "MIP3a (t-map) on *fap52*".

“All of the protofilaments in the B-tubules were completely missing in some DMTs, whereas DMTs remaining several protofilaments in the B-tubules were also observed.”

Please provide numbers (%).

We added the information in the result section; " ... revealed that the B-tubules were missing in ~33.8% of the outer DMTs (Fig. 4c). Among them, 23% lacked all of the protofilaments in the B-tubules, whereas 77% retained several protofilaments." (page 10)

*“...we also found that the density of dynein b, a species of inner arm dyneins was decreased in *fap45* and *fap52* (Supplementary Fig. S3a-c).”*

Earlier it was stated that mutants had a normal level of axonemal dyneins. I would suggest showing images of 96-nm repeats as in the main figure, describe the reduced level of dynein b and next changes in B-tubule luminal proteins.

Additionally, localization of the BCCP tagged FAP45 and FAP52 can be analyzed by cryo-ET.

In the new manuscript we mentioned the reduction of the dynein b density (page 8-9). As

mentioned above, we had difficulty to amplify the BCCP tag in the mutants. We also tried cryo-ET with sarcosyl treated axonemes, but we failed to acquire images because those axonemes could not maintain 9+2 and fell apart in the ice.

Data presented in FigS5 should be described in the Results section

Supplementary Figure S5 is Figure S8 in the new manuscript. We mentioned the figure in the result section; "These data strongly suggest that FAP45 and FAP52 stabilize the doublet B-tubule and prevent depolymerization induced by damage to the MT wall. However, the DMT of *fap45fap52* are still more stable than cytoplasmic microtubules, suggesting that B-tubules are also stabilized by other mechanisms, such as post-translational modifications of tubulin^{36, 37} (Supplementary Fig. S8) and fMIPs that bind inside the B-tubule along its length³⁵." (page 12).

"This suggests that B-tubules are also stabilized by other mechanisms, such as tubulin acetylation 32"

Please cite other REF

In addition to the reference, we cited 37. Xu, Z. *et al.* Microtubules acquire resistance from mechanical breakage through intraluminal acetylation. *Science* **356**, 328-332 (2017).

*Please, discuss data presented by:
Kirima and Oiwa, 2017, Cell Struct Funct
Stoddard et al., 2018, Mol Biol Cell*

We added discussion on the papers in the last paragraph of the discussion section; "The structures of MIPs in the A-tubule is more complex than in the B-tubule. It is previously shown that hyper-stable "ribbons" were composed of four protofilaments to which MIP4 is bound⁴⁰, suggesting MIP4 stabilizes the ribbon structure. A recent paper reported that FAP85 is a MIP in the A-tubule and stabilizes cytoplasmic microtubules in vitro²². Besides, a study using *Tetrahymena* revealed that Rib72A and B are essential for assembly of MIP6, which is important for proper flagellar beating²³. Although how these newly identified MIPs contribute to the stability of the A-tubule is still unclear, MIPs in A-tubules might also have fail-safe mechanisms, similar to MIP3a and c." (page 14)

Please provide:

1. Silver-stained gel and western blot of the purified proteins used to produce antibodies.

We added the images to Supplementary Figure S2a.

2. Silver-stained gel of Fap45 and Fap 52 precipitates

We only have the silver-stained gel of FAP45-IP. We added the image to Supplementary Figure S1d.

How many ug of the proteins was loaded on the gel (western bots)

5ug of isolated axonemes. We added the information to the methods (page 16).

Table S1 is showing only selected proteins and thus, it is difficult to make one's own mind about these results. Can you include also proteins identified by the lower number of peptides?

We extended the list to show proteins over 10 peptides were identified in KCl extracted axonemes.

Table S2 is showing peptides of only two proteins. Please show all proteins (including their MW) identified by mass spectrometry. Was corresponding fragment of the gel (control) analyzed by

mass spec? Any proteins in the control?

We showed all the peptides from *Chlamydomonas* we identified. We cut out the band from the gel shown in Supplementary Fig. 1d (arrowhead), and analyzed by mass spec.

Reviewer 3

The experiment relies on the concept that the doublet microtubule is attached with a very precise geometric constraint to the surface with the A-tubule immobilized and the B-tubule standing right up facing the HS-AFM tip. This is of such crucial importance for the entire conclusions that it must be precisely described and also conclusively documented in the main text.

We appreciate the important comment on how DMTs look like in HS-AFM observations. We observed different orientations of DMTs on the mica surface, which can be classified into three typical orientations as shown in Supplementary Figure S7. We described the classifications in the main text. "When DMTs are adsorbed on the mica surface, AFM images were classified into three types as shown in Supplementary Fig. S7; Class 1. The 24 nm-periodicity of ODAs was visualized on the top of the DMT, suggesting that the B-tubule and the radial spokes were immobilized on the mica; Class 2. Heads of the radial spokes in 96 nm-periodicity were visualized on the top of DMT, suggesting that both of the A and B-tubule were immobilized on the mica (Supplementary Fig. S7b); Class 3. The radial spokes are periodically and horizontally projected from the A-tubule, suggesting that the B-tubule was visualized on the top while the A-tubule was immobilized on the mica (Supplementary Fig. S7c)." (page 11) We selected the class 2 and 3 for the observation and mentioned it in the main text (page 11).

The progression of the defect is a qualitative measure. It will in HS-AFM depend on many parameters, such as tip geometry, scan direction and force.

2.1 Comment on the tips used and whether the experiments are performed with the same or essentially the same tip, and how this is controlled.

We appreciate the critical comment on the tip condition. As the reviewer pointed out, the progression of the defect should depend on various imaging parameters especially tip conditions. In general, it is hard to keep using an identical tip for all experiments because the tip end is easily contaminated or damaged. Therefore, we needed to exchange a tip in each experiment. However we selected images with similar quality (the width of MTs and spatial resolution) to avoid the possible tip effect and thus we believe that different tips do not affect defect progression so much. We added this discussion in the methods section of the main text (page 20).

2.2 The authors mainly (only) present results where the MT is oriented in line with the fast scan axis (horizontal in the image). Were the experiments also performed on differently oriented MTs, was the depolymerization then different, what were accepted angular deviations from the horizontal axis, etc?

As the reviewer pointed out, the relative scanning direction of the AFM tip could influence fragility of DMTs. Based on our experience of MT-observation by HS-AFM, MTs that oriented perpendicular to the fast scanning direction are more easily damaged by lateral force from the AFM tip than horizontally oriented MTs. Although DMTs are more stable and not easily damaged by the AFM tip even if they are immobilized with the vertical angle to the scan axis, we selected DMTs of which the angle to the fast-scan axis is less than 45 degrees to minimize accidental damage on the DMTs (we added this information to the method). Indeed, wild type DMTs angled with ~45 degrees to the fast-scan axis showed similar depolymerization rate to horizontal ones (Reviewer Movie 1).

2.3 The scanning force is obviously the most important parameter, as the depolymerization is

*induced by a force induced defect it is likely dependent on force. The applied force in a HS-AFM depends on the free oscillation amplitude $A(\text{free})$ and the setpoint amplitude $A(\text{set})$. The authors do not comment in detail on this. It has recently been shown (Miyagi, RSI, 2016) that both $A(\text{free})$ and $A(\text{set})$ can and must be controlled to control the applied tip-surface interaction. In the same work it has been shown that this is crucial for the lifetime of weak intermolecular interactions. Thus, the author must detail $A(\text{free})$ and $A(\text{set})$ and how $A(\text{free})$ was stabilized over extended experimental periods. Obviously, $A(\text{free})$ and $A(\text{set})$ must be quasi identical in all experiments (WT, *fap45*, *fap52*, *fap45fap52*) to be compared.*

We apologize that we did not describe detailed experimental conditions. We used the same amplitude of about 2.0 nm for a free oscillation and set the reduced amplitude of 1.6 nm for the feedback control. We confirmed that the optical-beam-deflection sensitivity was almost constant less than 5 % deviations for all experiments. The tapping force applying the sample was 120 pN from the simple equation proposed by Rodríguez et al. We added the experimental conditions of tapping-mode imaging in the methods section (page 20).

We did not employ the drift compensation as proposed by Miyagi *et al* in this experiment because we imaged MTs for a relatively short duration less than 1 min. This short duration time of the imaging usually does not change the free oscillation amplitude. Actually, we confirmed that the free oscillation amplitude was not changed before and after the imaging. Thus, the tip-surface force should be almost constant during the imaging.

1 The false color scale for the representation of the HS-AFM data is an unusual and inappropriate choice (Figure 4). While the supplementary movies are very clear, the figure displays the highest protruding structures in a teal tone that is not intuitively associated with stronger protrusion than the green and yellow. It almost looks like a contrast inversion and when printed in gray scales all these colors have the same intensity.

We changed the color of the snap-shots same as the supplementary movies.

2 Plotting the progression of depolymerization towards proximal and distal directions would be nice. Why is there no difference?

As the reviewer suggested, microtubules have polarity and depolymerization from the plus end toward the minus end is faster than the opposite in catastrophe of MTs. However, we did not observe significant difference between both directions in *Chlamydomonas* DMTs. This might be because the DMTs are also stabilized by acetylation and fMIPs found in Ichikawa et al., 2017.

The mention of FAP20 occurs once, and it is unclear how it relates to the other findings.

FAP20 and FAP52 cooperatively reinforce the inner junction. The lack of both proteins is required for the detachment of the B-tubule from the A-tubule.

The sentence "recent studies revealed that the assembly of cilia is significantly decreased in several types of tumors^{5,6}, suggesting a close relationship between cilia and tumorigenesis" seems like an overstatement. There are many things different in tumors, and thus "close relationship" sounds like a strong call.

As the reviewer pointed out, relation between cilia and tumorigenesis is still controversial and too few have revealed to say "a close relationship". It is possible that the loss of cilia is just a consequence of tumorigenesis. Therefore, we rewrote the sentence; "recent studies revealed that the assembly of cilia is significantly decreased in several types of tumors^{5, 6}, implying some correlation between ciliogenesis and tumorigenesis." (page 3)

Figure1:

Coloring the different elements in Figure1a would be beneficial for the understanding.

We colored major structures in Figure 1a.

"Recent advances in cryo-ET techniques have dramatically revealed structural details" - delete "dramatically".

We deleted "dramatically".

"To reveal the defect caused by the loss of FAP45 and FAP52, we observed the mutant axonemes by cryo-ET. Interestingly, the mutants had structural defects inside of the B-tubule (Fig. 2)." - "In agreement" rather than "Interestingly".

We rewrote the sentence; " In agreement with the above results, the mutants also showed structural defects inside of the B-tubule (Fig. 3)." (page 9)

Reviewers' Comments:

Reviewer #1:

Remarks to the Author:

I really appreciate that the authors have made tremendous efforts to improve the manuscript and address the reviewers' concerns and suggestions. The readability of the manuscript is much better and is more appropriate for publication. I still have a few comments:

I am convinced that FAP45 is the filamentous MIP running in between B8 and B9. However, with the resolution from this study, it is unlikely that the missing density of MIP3c is the density of fMIP. In the other region of the doublet, the surface inside looks also smooth but there should still be fMIP there. It is likely that the obvious density of MIP3c from Student-t test difference map is from other unidentified MIPs binding on top of FAP45. Is there any evidence of additional proteins also missing in the FAP45 mutant?

Page 8: This result suggests that the number of dynein b may be reduced or dynein b to the B-tubule in the mutants may be more flexible than in wild type.

I think you mean "may be reduced or the binding of dynein b to the B-tubule ...". Also, does the author have any structural evidence that dynein b bind to B-tubule? It doesn't look like to me that dynein b is binding to B-tubule. It can be other reason such as the flagella in the mutant is shorter and there is a difference between the proximal and distal region.

The point about confluent medium is more viscous was raised by reviewer 2. Though the author change the text, I still wonder is there any evidence that confluent medium is more viscous?

Supplementary Figure 5

The author claim a FSC threshold at 0.143 of 2.7-2.8 nm. But it is not clear that the FSC is a Gold Standard FSC or a FSC from two halves that are refined using the same reference. If it is not Gold-standard FSC, the resolution should be reported at 0.5 threshold.

Reviewer #2:

Remarks to the Author:

All issues were addressed. I congratulate the Authors and I am happy to recommend this manuscript for publication.

Below are very minor suggestions to the text:

Page 4:

"Recent advances in cryo-ET techniques have revealed structural details of DMTs, yet a significant question remains—What stabilizes DMTs?" (...question remains — what stabilizes DMTs?)

Page 6:

"Therefore, we hypothesized that FAP45 and FAP52 proteins play important roles in stabilizing DMTs".

Suggestion:

"We hypothesized that FAP45 and FAP52 proteins may play a roles in stabilizing DMTs".

Pages 6-7:

"Despite FAP45 and FAP52 being crosslinked by a zero-length crosslinker, they localized on the axoneme independent of each other (Fig. 1b). Consistent with the mass spectroscopic

analysis, the ~130 kDa crosslinked product of FAP45 and FAP52 was not detected in the crosslinked fap45 or fap52 axoneme (Fig. 1c and d, filled arrowheads), suggesting that FAP45 and FAP52 are neighbors on the axoneme."

Suggestion:

The ~130 kDa product of FAP45 and FAP52 crosslinking was not detected in the fap45 or fap52 axoneme exposed to a zero-length crosslinker (Fig. 1c and d, filled arrowheads). Thus, likely, FAP45 and FAP52 are neighbors on the axoneme but localized independent of each other (Fig. 1b).

Page 8:

"This result suggests that the number of dynein b may be reduced or dynein b to the B-tubule in the mutants may be more flexible than in wild type."

Suggestion:

"This result suggests that in the mutants the number of dynein b may be reduced or that dynein b... ..to the B-tubule may be more flexible than in wild type."

Page 9:

"A comparison between the wild type and fap52 DMT revealed an arch-like density is missing in the B-tubule of the fap52 mutant"

Suggestion:

"A comparison between the wild type and fap52 DMT revealed that an arch-like density is missing in the B-tubule of the fap52 mutant"

Page 15:

"Although how these newly identified MIPs contribute to the stability of the A-tubule is still unclear, MIPs in A-tubules might also have fail-safe mechanisms, similar to MIP3a and c."

Suggestion:

"Although how these newly identified MIPs contribute to the stability of the A-tubule is still unclear. It is possible that MIPs in A-tubules might also have fail-safe mechanisms, similar to MIP3a and c."

Reviewer #3:

Remarks to the Author:

I thank the authors for their thoughtful and nice revision.

The paper is now acceptable for publication.

I wished however the authors to do one last amendment:

In their response to my comment about the scanning force (I think this is an important point and particularly important for this study where the authors make use of the force to manipulate the sample), the authors say that they refer to Rodriguez et al. 2003. But instead they refer to Uchihashi et al 2011 (ref 52). Please change ref 52 to Rodriguez et al. 2003 and indicate the equation used to estimate the force (I guess it is equation 10).

Response to reviewers' comments

Reviewer #1

I am convinced that FAP45 is the filamentous MIP running in between B8 and B9. However, with the resolution from this study, it is unlikely that the missing density of MIP3c is the density of fMIP. In the other region of the doublet, the surface inside looks also smooth but there should still be fMIP there. It is likely that the obvious density of MIP3c from Student-t test difference map is from other unidentified MIPs binding on top of FAP45. Is there any evidence of additional proteins also missing in the FAP45 mutant?

We did not identify proteins missing in the *fap45* axoneme other than FAP45. We consider that MIP3c is a lateral filament, which connects MIP3a and b to fMIPs. Similar densities were observed in the *Tetrahymena* sub-nanometer resolution, so we rewrote that part of the main text; " These densities are arranged in a 48-nm repeat, and reminiscent of a lateral filament connecting MIP3, fMIP B8B9, and fMIP B7B8 in *Tetrahymena* DMTs³⁵." (page 9)

Page 8: This result suggests that the number of dynein b may be reduced or dynein b to the B-tubule in the mutants may be more flexible than in wild type.

I think you mean "may be reduced or the binding of dynein b to the B-tubule ...". Also, does the author have any structural evidence that dynein b bind to B-tubule? It doesn't look like to me that dynein b is binding to B-tubule. It can be other reason such as the flagella in the mutant is shorter and there is a difference between the proximal and distal region.

We also don't think that dynein b is binding to the B-tubule in the mutants. The flagellar length of the mutants is normal, so that the reduction of the dynein b density is not due to the difference between the proximal and distal. Therefore, we rewrote the sentence; " This result suggests that in the mutants the number of dynein b may be reduced or dynein b cannot bind to the B-tubule properly for some reason."

The point about confluent medium is more viscous was raised by reviewer 2. Though the author change the text, I still wonder is there any evidence that confluent medium is more viscous?

Since we did not find evidence for the speculation, we changed the text in the previous revision.

Supplementary Figure 5

The authors claim a FSC threshold at 0.143 of 2.7-2.8 nm. But it is not clear that the FSC is a Gold Standard FSC or a FSC from two halves that are refined using the same reference. If it is not Gold-standard FSC, the resolution should be reported at 0.5 threshold.

We re-calculated FSC using the gold-standard method and replaced the figure (Supplementary Figure 5f). Also, we described the method in the legend and the effective resolutions (wild type and *fap52*: 2.9 nm; *fap45*: 2.8 nm).

Reviewer #2

Page 4:

"Recent advances in cryo-ET techniques have revealed structural details of DMTs, yet a significant question remains—What stabilizes DMTs?" (...question remains — what stabilizes DMTs?)

The suggestion has been reflected in the main text.

Page 6:

"Therefore, we hypothesized that FAP45 and FAP52 proteins play important roles in stabilizing DMTs".

Suggestion:

"We hypothesized that FAP45 and FAP52 proteins may play a roles in stabilizing DMTs".

We rewrote the sentence; " We hypothesized that FAP45 and FAP52 proteins play a role in stabilizing DMTs."

Pages 6-7:

"Despite FAP45 and FAP52 being crosslinked by a zero-length crosslinker, they localized on the axoneme independent of each other (Fig. 1b). Consistent with the mass spectroscopic analysis, the ~130 kDa crosslinked product of FAP45 and FAP52 was not detected in the crosslinked *fab45* or *fab52* axoneme (Fig. 1c and d, filled arrowheads), suggesting that FAP45

and FAP52 are neighbors on the axoneme."

Suggestion:

The ~130 kDa product of FAP45 and FAP52 crosslinking was not detected in the *fab45* or *fab52* axoneme exposed to a zero-length crosslinker (Fig. 1c and d, filled arrowheads). Thus, likely, FAP45 and FAP52 are neighbors on the axoneme but localized independent of each other (Fig. 1b).

We apologize that the paragraph was confusing. The crosslinking result just tells consistency of the mass spec results. So, we rewrote the paragraph; " On the other hand, *fab45* and *fab52* axonemes retained known major axonemal components (Fig. 1b). Also, FAP45 and FAP52 were incorporated into the axoneme independently of each other (Fig. 1b). The ~130 kDa crosslinked product of FAP45 and FAP52 was detected in the wild type axoneme treated with EDC but not in the crosslinked *fab45* or *fab52* axoneme (Fig. 1c and d, filled arrowheads), consistent with the mass spectroscopic analysis."

Page 8:

"This result suggests that the number of dynein b may be reduced or dynein b to the B-tubule in the mutants may be more flexible than in wild type."

Suggestion:

"This result suggests that in the mutants the number of dynein b may be reduced or that dynein b..to the B-tubule may be more flexible than in wild type."

We rewrote the sentence; " This result suggests that in the mutants the number of dynein b may be reduced or dynein b cannot bind to the B-tubule properly for some reason."

Page 9:

"A comparison between the wild type and fap52 DMT revealed an arch-like density is missing in the B-tubule of the fap52 mutant"

Suggestion:

"A comparison between the wild type and fap52 DMT revealed that an arch-like density is missing in the B-tubule of the fap52 mutant"

The suggestion has been reflected in the main text.

Page 15:

"Although how these newly identified MIPs contribute to the stability of the A-tubule is still unclear, MIPs in A-tubules might also have fail-safe mechanisms, similar to MIP3a and c."

Suggestion:

"Although how these newly identified MIPs contribute to the stability of the A-tubule is still unclear. It is possible that MIPs in A-tubules might also have fail-safe mechanisms, similar to MIP3a and c."

We rewrote the sentence; " Although how these newly identified MIPs contribute to the stability of the A-tubule is still unclear, it is possible that MIPs in A-tubules also have fail-safe mechanisms, similar to MIP3a and c."

Reviewer #3

In their response to my comment about the scanning force (I think this is an important point and particularly important for this study where the authors make use of the force to manipulate the sample), the authors say that they refer to Rodriguez et al. 2003. But instead they refer to Uchihashi et al 2011 (ref 52). Please change ref 52 to Rodriguez et al. 2003 and indicate the equation used to estimate the force (I guess it is equation 10).

We changed the reference to Rodriguez et al., 2003. As the reviewer expected, we used the equation 10 for the estimation and described it in the methods section.